# Efficient Autoregressive Inference for Transformer Probabilistic Models

**Conor Hassan**[1,2]*, **Nasrulloh Loka**[3]*, **Cen-You Li**[3], **Daolang Huang**[1,2], **Paul E. Chang**[3,4],
**Yang Yang**[3], **Francesco Silvestrin**[3], **Samuel Kaski**[1,2,5], **Luigi Acerbi**[3]

[1]Department of Computer Science, Aalto University, Finland     [2]ELLIS Institute Finland
[3]Department of Computer Science, University of Helsinki, Finland     [4]Verda
[5]Department of Computer Science, University of Manchester, UK
`conor.hassan@aalto.fi`

## Abstract

Set-based transformer models for amortized probabilistic inference and meta-learning, such as neural processes, prior-fitted networks, and tabular foundation models, excel at single-pass *marginal* prediction. However, many applications require *joint distributions* over multiple predictions. Purely autoregressive architectures generate these efficiently but sacrifice flexible set-conditioning. Obtaining joint distributions from set-based models requires re-encoding the entire context at each autoregressive step, which scales poorly. We introduce a *causal autoregressive buffer* that combines the strengths of both paradigms. The model encodes the context once and caches it; a lightweight causal buffer captures dependencies among generated targets, with each new prediction attending to both the cached context and all previously predicted targets added to the buffer. This enables efficient batched autoregressive sampling and joint predictive density evaluation. Training integrates set-based and autoregressive modes through masked attention at minimal overhead. Across synthetic functions, EEG time series, a Bayesian model comparison task, and tabular regression, our method closely matches the performance of full context re-encoding while delivering up to $20\times$ faster joint sampling and density evaluation, and up to $7\times$ lower memory usage.

## 1 Introduction

Recent advances in amortized probabilistic inference and meta-learning have produced set-based conditioning models that rapidly adapt to new tasks without retraining. Prominent examples include *neural processes* (NPs; Garnelo et al. 2018a; Foong et al. 2020), their transformer-based extensions (Nguyen & Grover, 2022; Chang et al., 2025), *prior-fitted networks* (PFNs; Müller et al. 2022), and *tabular foundation models* (Hollmann et al., 2023; 2025; Qu et al., 2025; 2026; Zhang et al., 2025). These methods share a key architectural principle: they process variable-sized *context sets* through permutation-invariant encoders that respect the exchangeability of observed data, enabling accurate marginal predictive distributions over new target variables in a single forward pass.

While these models are highly efficient for *marginal* predictions, many applications require coherent *joint* distributions over multiple targets. Tasks such as signal interpolation, behavioral data modeling, and multi-column tabular prediction demand capturing dependencies across target variables. The standard approach deploys these models autoregressively at test time (AR; Bruinsma et al. 2023): to generate $K$ predictions, the $k$-th step conditions on the initial context $\mathcal{C}$ plus all $k-1$ previous predictions. Since the permutation-invariant encoder uses bidirectional self-attention over all inputs, each appended prediction triggers a complete re-encoding of the entire augmented set, yielding $\mathcal{O}(K(N+K)^2)$ complexity. This severely limits applications with large contexts ($N$), long target sequences ($K$), or frequent sampling requirements. Advances in efficient attention (Jaegle et al., 2021; Feng et al., 2023) can reduce costs for encoding large *static* contexts but do not address the repeated recomputation inherent in autoregressive expansion of the conditioning set.

---

*Shared first co-authorship

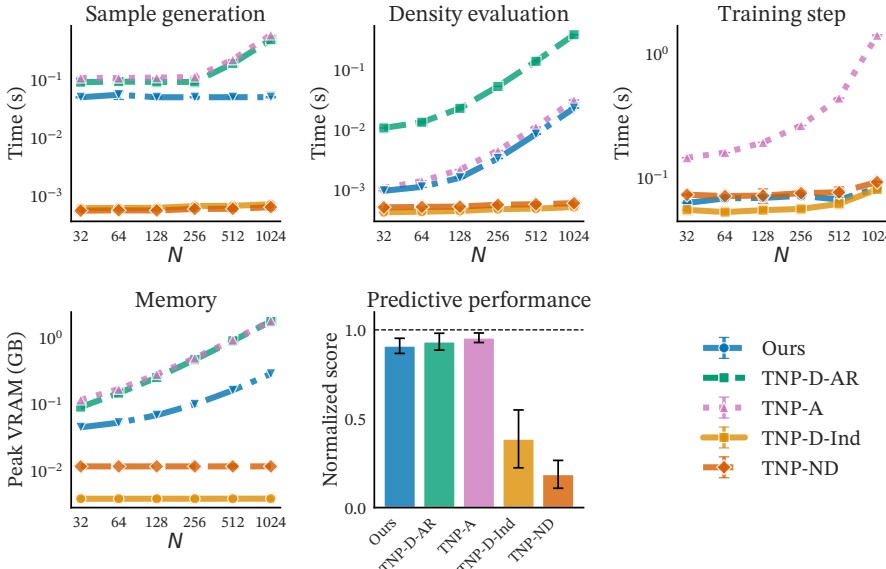

Figure 1: Our method closely matches the predictive performance of expensive joint-prediction methods while delivering significant speedups and lower memory usage. We compare against transformer-based probabilistic models spanning the efficiency-expressivity tradeoff (see Section 5 for details). Our approach reduces the cost of autoregressive joint inference from $\mathcal{O}(K(N+K)^2)$ to $\mathcal{O}(N^2+NK+K^2)$, yielding up to $20\times$ faster sampling and density evaluation, and $7\times$ lower peak memory. Panels show wall-clock time (log scale) for **(a)** sampling (top left), **(b)** density evaluation (top middle), **(c)** a training step (top right), **(d)** peak memory usage versus context points $N$ (bottom left), and **(e)** normalized predictive performance averaged across six tasks (bottom middle).

To address this limitation, we introduce the *causal autoregressive buffer*, an architectural mechanism that decouples the expensive encoding of the static context from lightweight sequential prediction. Inspired by the efficiency of purely autoregressive architectures in language modeling (Brown et al., 2020) and image generation (Chen et al., 2020; Li et al., 2024), our buffer implements a causal attention pattern for managing dependencies among generated targets – but crucially, it operates alongside the set-based context rather than replacing it. Our approach encodes the initial context $\mathcal{C}$ once and caches its representation. Targets added to the buffer attend to both the static context cache and previously predicted targets through causal masking, capturing dependencies among newly generated samples without requiring context re-encoding, reducing cost from $\mathcal{O}(K(N+K)^2)$ to $\mathcal{O}(N^2+NK+K^2)$. When the buffer is empty, the model reduces to a standard set-based predictor, preserving marginal prediction quality. A unified training strategy using masked attention and a buffer-size curriculum allows a single model to handle both efficient marginal predictions and accelerated autoregressive sampling and joint predictive density evaluation with substantial speedups and comparable predictive accuracy to standard AR deployment (Fig. 1).

Our main contributions are:

1. We introduce the *causal autoregressive buffer*, which reduces the cost of autoregressive joint inference for transformer-based amortized probabilistic models from $\mathcal{O}(K(N+K)^2)$ to $\mathcal{O}(N^2+NK+K^2)$ by decoupling one-time context encoding from sequential prediction, enabling both efficient autoregressive sampling and one-pass joint predictive density evaluation – while recovering exact standard-model behavior when the buffer is empty.

2. We propose a unified training strategy, combining masked attention with a buffer-size curriculum, that enables a single model to learn both modes of operation at minimal cost.

3. We demonstrate applicability across transformer-based probabilistic models including TNPs/PFNs (Nguyen & Grover, 2022; Müller et al., 2022) and tabular foundation models (TabICL; Qu et al., 2025), achieving up to $20\times$ faster joint sampling and $7\times$ lower memory while matching the predictive accuracy of full context re-encoding across tasks.

## 2 PRELIMINARIES

We consider the task of predicting outputs at new inputs, given a set of observed input-output pairs. Given a *context set* $\mathcal{C} = \{(\mathbf{x}_n, y_n)\}_{n=1}^N$ with $N$ input-output pairs and a *target set* $\mathcal{T} = \{(\mathbf{x}_m^\star, y_m^\star)\}_{m=1}^M$, we seek to learn a predictive distribution $p_{\boldsymbol{\theta}}(y_{1:M}^\star \mid \mathbf{x}_{1:M}^\star; \mathcal{C})$ where $\boldsymbol{\theta}$ are the model's learnable parameters (Foong et al., 2020). This setup arises in meta-learning, few-shot prediction, and tabular foundation models – any setting where a single model must condition on variable-sized observed data without task-specific retraining.

**Transformer diagonal prediction maps.** Transformer architectures (Vaswani et al., 2017) are a natural fit for this set-based task. Methods such as (diagonal) *transformer neural processes* (TNPs; Nguyen & Grover, 2022) and *prior-fitted networks* (PFNs; Müller et al., 2022) use two core attention mechanisms. First, the model processes $\mathcal{C}$ using multi-head self-attention (MHSA). Then, each target input $\mathbf{x}_m^\star$ queries this summary using multi-head cross-attention (MHCA). This structure leads to an efficient *diagonal* predictive model where predictions are conditionally independent:

$$p_{\boldsymbol{\theta}}(y_{1:M}^\star \mid \mathbf{x}_{1:M}^\star; \mathcal{C}) = \prod_{m=1}^M p_{\boldsymbol{\theta}}(y_m^\star \mid \mathbf{r}_{\text{tgt}}(\mathbf{x}_m^\star, \mathbf{r}_{\mathcal{C}}(\mathcal{C}))). \tag{1}$$

Here, $\mathbf{r}_{\mathcal{C}}(\mathcal{C})$ is the permutation-invariant summary of the context produced by the MHSA layers[1], and $\mathbf{r}_{\text{tgt}}(\cdot, \cdot)$ is the final decoding function that produces a parametric prediction for $y_m^\star$ via MHCA. This may consist of a single Gaussian, but more expressive parameterizations include Riemannian distributions (Müller et al., 2022) and mixtures of Gaussians (Uria et al., 2016; Chang et al., 2025). These models are efficiently trained via maximum-likelihood on random context-targets data splits.

**Autoregressive sampling and predictive density evaluation.** Many applications require joint distributions over targets, both for *generating coherent joint samples* and for *evaluating joint predictive densities*. While Eq. (1) can be extended with parametric multivariate densities such as a full Gaussian (Markou et al., 2022; Nguyen & Grover, 2022), a more flexible alternative applies an autoregressive factorization of the same model (Bruinsma et al., 2023). We switch from index $m$ to $k$ (and $M$ to $K$) to emphasize that targets are now processed in a specific sequential order:

$$p_{\boldsymbol{\theta}}(y_{1:K}^\star \mid \mathbf{x}_{1:K}^\star; \mathcal{C}) = \prod_{k=1}^K p_{\boldsymbol{\theta}}\left(y_k^\star \mid \mathbf{x}_k^\star; \mathcal{C} \cup \{(\mathbf{x}_j^\star, y_j^\star)\}_{j=1}^{k-1}\right). \tag{2}$$

Crucially, Eq. (2) does not define a new model; it is a *mode of deployment* for models described by Eq. (1).[2] Each prediction is conditioned on all previously predicted targets, capturing inter-target dependencies. This flexibility comes at a cost: because the permutation-invariant encoder uses bidirectional self-attention, adding a single predicted target to the conditioning set invalidates all cached representations, forcing a complete re-encoding of the augmented set. The resulting complexity is $\mathcal{O}(K(N+K)^2)$, and generating $B$ parallel sample sequences requires $B$ independent forward-pass streams, each maintaining its own growing context.

Our goal is to improve efficiency for both sequential and parallel sampling and predictive density evaluation by encoding the context once and reusing it throughout. Existing autoregressive update schemes break this caching: when targets join the conditioning set, the context representation must be recomputed. Our key insight is to separate the roles of initial context $\mathcal{C}$ and predicted targets $\{(\mathbf{x}_j^\star, y_j^\star)\}_{j<k}$. We preserve permutation invariance for the initial context (encoded once and cached) while handling target dependencies through a separate causal mechanism. When needed, the buffer can be merged back into the context to restore full permutation invariance. This selective relaxation – in-between fully set-based and purely autoregressive models – enables efficient sequential and parallel operations while maintaining the strengths of set-based conditioning.

---

[1] See Appendix H.1 for evidence of the impact of permutation invariance for the context set.

[2] Eq. (2) imposes a specific factorization order. While fixing the order can be a valid modeling choice under certain circumstances, this breaks permutation invariance. In cases where permutation invariance is required, a Monte Carlo approximation can be obtained by averaging over multiple target orderings.

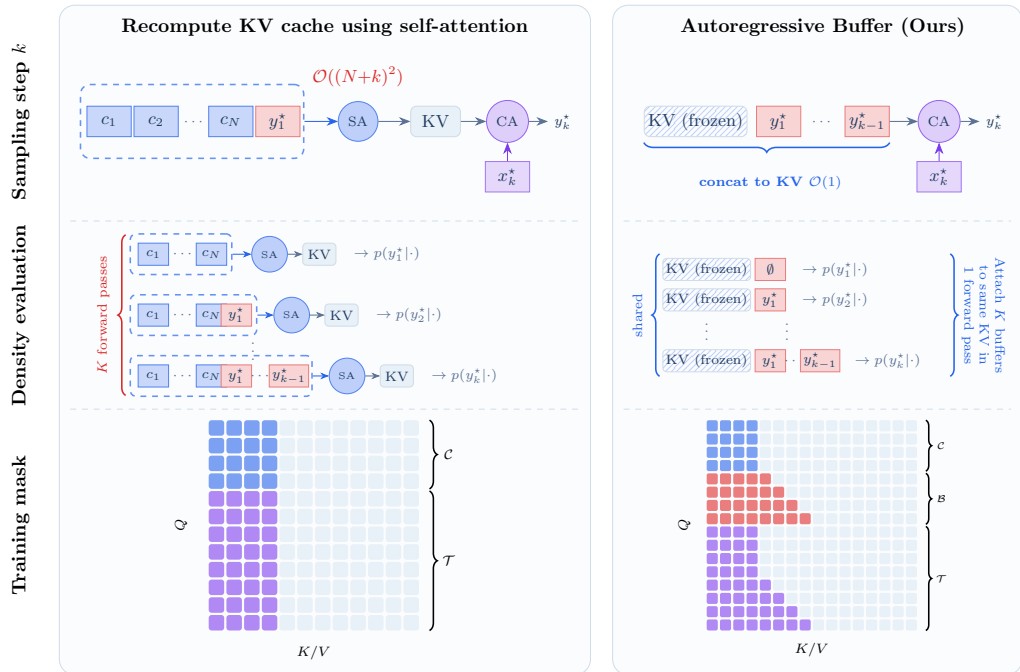

Figure 2: Standard autoregressive (AR) method compared to the proposed causal AR buffer. **Top (sampling):** Standard AR (left) appends each new prediction $(\mathbf{x}_k^\star, y_k^\star)$ to the context set and re-computes self-attention (SA) over the growing set at each step, with per-step cost $\mathcal{O}\big((N + k)^2\big)$. Our causal AR buffer (right) caches the context KV once and appends each new buffer token to the frozen KV cache, reducing attention cost. **Middle (density evaluation):** Standard AR requires $K$ forward passes to compute the conditionals $p(y_k^\star \mid \cdot)$ because the context set changes at each step (left), whereas the AR buffers (right) enables computing all $K$ conditionals in a single pass. **Bottom (training mask):** (left) Standard AR training mask: context self-attention (blue) and target-to-context attention (purple); (right) our training mask: buffer tokens attend to the context and to one another causally (red). Target tokens are either attend only to the context, or to the context plus a variable number of buffer tokens (purple).

## 3 EFFICIENT AUTOREGRESSIVE INFERENCE

**Core contribution.** Our method conditions predictions on a static, task-defining *context* $\mathcal{C}$ and a dynamic *autoregressive buffer* $\mathcal{B}$ (see Fig. 2). We parameterize the predictive distribution as

$$p_{\boldsymbol{\theta}}(y_{1:K}^\star \mid \mathbf{x}_{1:K}^\star; \mathcal{C}) = \prod_{k=1}^{K} p_{\boldsymbol{\theta}}\big(y_k^\star \mid \mathbf{r}_{\text{tgt}}(\mathbf{x}_k^\star, [\mathbf{r}_{\mathcal{C}}(\mathcal{C}), \mathbf{b}_{1:k-1}])\big), \quad \mathbf{b}_k = \mathbf{r}_{\mathcal{B}}\big((\mathbf{x}_k^\star, y_k^\star), [\mathbf{r}_{\mathcal{C}}(\mathcal{C}), \mathbf{b}_{1:k-1}]\big),$$

(3)

where $\mathbf{r}_{\mathcal{B}}$ is the buffer encoder, implemented via causal MHSA among buffer tokens and cross-attention to the cached context at each layer; $\mathbf{b}_{1:k}$ denotes the first $k$ encoded buffer representations; and $\mathbf{b}_{1:0} = \emptyset$. Crucially, $\mathbf{r}_{\mathcal{C}}(\mathcal{C})$ is computed once and cached. The target decoder $\mathbf{r}_{\text{tgt}}$ cross-attends to the concatenated keys/values from the cached context and the visible buffer prefix, then passes the result through a distribution head (e.g., an MLP parameterizing a mixture of Gaussians).[3] When the buffer is empty, Eq. (3) reduces to the standard diagonal prediction map (Eq. (1)).

To couple one-time set-based encoding with sequential dependence, the attention must satisfy four requirements: **(R1)** the *context is immutable*: encoded once and cached as read-only; **(R2)** the *buffer*

---

[3]$K$ denotes the total number of AR targets; the buffer stores up to $K-1$ previous predictions. Thus, "buffer size $K=16$" means at most 15 buffered elements and 16 total predictions. $K=1$ corresponds to standard AR inference (empty buffer).

*is strictly causal*: buffer token $j$ may attend only to the cached context and positions $< j$; **(R3)** *information flows out of the context but never back*: no edges write into $\mathcal{C}$; and **(R4)** each *target* attends to the cached context and *visible buffer prefix*; targets do not attend to one another.

During training, we enforce **(R1)**–**(R4)** in a single forward pass using a structured attention mask. We implement this with a transformer backbone that processes context, buffer, and target tokens with distinct role embeddings; buffer tokens carry learned positional embeddings indicating their autoregressive order (see Appendix H.2). This enables parallel loss computation by conditioning each target's prediction on the context and a variable-sized, ground-truth buffer prefix. *At inference*, we use a two-stage process: a one-time context encoding followed by *prediction* as either *sampling* or *predictive density evaluation*. The total attention cost is $\mathcal{O}(N^2 + KN + K^2)$: a one-time $\mathcal{O}(N^2)$ for context self-attention, $\mathcal{O}(KN)$ for cross-attention reads from the cache, and $\mathcal{O}(K^2)$ for causal buffer self-attention and target-to-buffer reads. By contrast, naive autoregressive methods cost $\mathcal{O}(K(N+K)^2)$ due to repeated context recomputation. Architectural details appear in Appendix A.

**Training details.** The model is trained by minimizing the expected negative log-likelihood over a prior distribution of datasets $\mathcal{P}$. Each training task is generated by sampling a dataset $\mathcal{D} = \{(\mathbf{x}_i, y_i)\}_{i=1}^{N_{\text{tot}}} \sim \mathcal{P}$. We then split $\mathcal{D}$ into three disjoint sets using a random partition $\pi$: (1) the *context set* $\mathcal{C} = \{(\mathbf{x}_n, y_n)\}_{n=1}^{N}$; (2) the *buffer set* $\mathcal{B} = \{(\mathbf{x}_k, y_k)\}_{k=1}^{K}$; and (3) the *target set* $\mathcal{T} = \{(\mathbf{x}_m, y_m)\}_{m=1}^{M}$, with $N_{\text{tot}} = N + K + M$. We randomly order the buffer set $\mathcal{B}$ and compute all predictions for the target set $\mathcal{T}$ in a single forward pass. A structured attention mask controls whether each target can attend to the buffer, and if so, how many elements: 50% of the targets attend only to the context $\mathcal{C}$; 50% attend to the context plus a prefix of the buffer $\mathcal{B}_{1:v_m}$, where $v_m \sim \text{Uniform}\{1, \ldots, K\}$ (see Fig. 2, bottom). The training objective is:

$$\mathcal{L}(\boldsymbol{\theta}) = \mathbb{E}_{\mathcal{D}\sim\mathcal{P}}\left[\mathbb{E}_{(\mathcal{C},\mathcal{B},\mathcal{T})\sim\pi(\cdot|\mathcal{D})}\left[-\sum_{m=1}^{M}\log p_{\boldsymbol{\theta}}(y_m \mid \mathbf{x}_m, \mathcal{C}, \mathcal{B}_{1:v_m})\right]\right], \qquad (4)$$

where $\mathcal{B}_{1:v_m}$ is the visible portion of the buffer for target $m$ ($v_m = 0$ for context-only targets). This training curriculum ensures robust performance regardless of buffer state at test time: context-only targets maintain marginal prediction quality, while variable-length buffer exposure teaches the model to flexibly incorporate additional conditioning information. For any fixed conditioning set, minimizing the negative log-likelihood is equivalent to minimizing the KL divergence between the model and the true posterior predictive distribution (Müller et al., 2022; Elsemüller et al., 2024); our objective aggregates this across varying amounts of buffer information.

At inference, we support two modes: (i) *autoregressive sampling*, where the buffer grows incrementally with the model's own generated samples (rather than the ground-truth data used during training); and (ii) *parallel joint predictive density evaluation*, where we pack $K$ buffer tokens containing observed target values and $K$ query tokens to evaluate all $K$ conditionals in one pass (see below). The attention structure is identical in both inference modes; only the execution differs (prefill followed by sequential updates for sampling, single masked pass for evaluation).

**Autoregressive sampling.** Given a context $\mathcal{C}$ and a sequence of target inputs $\mathbf{x}_1^\star, \ldots, \mathbf{x}_K^\star$, we generate samples by first performing a one-time *prefill* of $\mathcal{C}$, caching its keys and values in an $\mathcal{O}(N^2)$ operation. We then *decode sequentially* following Eq. (3): for each step $k = 1, \ldots, K$, we form a target query for input $\mathbf{x}_k^\star$, attend to the cached context and causal buffer $\mathcal{B}_{k-1}$, sample $y_k^\star$ from the predictive distribution, and append $(\mathbf{x}_k^\star, y_k^\star)$ to the buffer with its positional embedding (see Fig. 2, top). Only the buffer's key/value cache is incrementally updated, avoiding context recomputation and yielding $\mathcal{O}(N^2 + NK + K^2)$ total complexity (detailed in Algorithm 1 in Appendix A.3).

**Joint predictive density evaluation.** Our framework also evaluates the joint predictive density of $K$ targets, $(\mathbf{x}_k^\star, y_k^\star)_{k=1}^{K}$, in a single forward pass. Following the TNP-A variant of Nguyen & Grover (2022), we pack two sets of tokens into the model: (i) *buffer tokens* $(\mathbf{x}_k^\star, y_k^\star)_{k=1}^{K}$, and (ii) *query tokens* $\mathbf{x}_k^\star{}_{k=1}^{K}$ for the same inputs. A causal attention mask ensures that each query for $\mathbf{x}_k^\star$ attends to the context $\mathcal{C}$ and only the preceding buffer tokens $\mathcal{B}_{1:k-1} = \{(\mathbf{x}_j^\star, y_j^\star)\}_{j<k}$, yielding:

$$\log p_{\boldsymbol{\theta}}(y_{1:K}^\star \mid \mathbf{x}_{1:K}^\star, \mathcal{C}) = \sum_{k=1}^{K}\log p_{\boldsymbol{\theta}}\big(y_k^\star \mid \mathbf{x}_k^\star, \mathcal{C}, \mathcal{B}_{1:k-1}\big). \qquad (5)$$

This yields the same joint log-density as sequential autoregressive evaluation but executes in a single forward pass with total attention cost $\mathcal{O}(N^2+KN+K^2)$ (see Fig. 2, middle). For a detailed description of the algorithm, see Algorithm 2 in Appendix A.3. Notably, autoregressive predictive density estimates are order-dependent; to recover approximate permutation invariance, we average the predictive densities over multiple buffer orderings (Murphy et al., 2019). See Appendix H.3 for an analysis of how the number of buffer orderings affects estimate stability.

**Batched autoregressive sampling.** Our method is particularly efficient for autoregressively generating multiple samples in a batch, conditioned on the same context $\mathcal{C}$ (e.g., multiple joint predictions for the same observed function values – see Fig. 2, top). A naive batched autoregressive approach must re-encode a growing context set at every generation step for each of the $B$ samples. To generate $B$ samples of length $K$, this results in a prohibitive total cost of $\mathcal{O}(BK(N + K)^2)$. In contrast, our approach performs the expensive context prefill ($\mathcal{O}(N^2)$) only *once*. This single context cache is then efficiently reused across all $B$ batched generation streams, with only the small, dynamic buffer maintaining a separate state for each sample. This reduces the total cost to $\mathcal{O}(N^2 + B(NK + K^2))$, making batched sampling practical even for large contexts and batches.

**Architectural generality.** Our buffer is a general mechanism applicable to other transformer variants. For instance, a Perceiver-style encoder (Jaegle et al., 2021) summarizes the context $\mathcal{C}$ into a fixed set of $P \ll N$ latent tokens, also known as *pseudo-tokens* (Lee et al., 2019; Feng et al., 2023; Lara-Rangel et al., 2025). We can precompute the latent key/value representations once – autoregressive decoding then requires attending only to these $P$ latents and the growing causal buffer. The per-layer attention cost is $\mathcal{O}(NP+P^2)$ for the prefill and $\mathcal{O}(PK+K^2)$ for decoding $K$ samples. Without our buffer, the cost is $\mathcal{O}(NPK+P^2K+PK^2)$. To demonstrate this, we applied our buffer to the *latent bottlenecked attentive neural process* (LBANP; Feng et al., 2023) architecture, a TNP variant that encodes the context with pseudo-tokens. We found that our buffer yields higher predictive densities than standard autoregressive inference with LBANP, likely because the buffer allows the model to condition on both the latent summary and the explicit history of previous points, rather than relying on the compressed latents alone. See Appendix H.4 for results and additional details.

## 4 RELATED WORK

**Transformer probabilistic models.** Our method serves as a modular component within neural processes (NPs; Garnelo et al., 2018b;a; Bruinsma et al., 2021; Nguyen & Grover, 2022; Dutordoir et al., 2023; Chang et al., 2025) and prior-fitted networks (PFNs; Müller et al., 2022; 2023; Hollmann et al., 2023). Prior work on efficient NP architectures has focused on improving scalability with respect to context set size (Feng et al., 2022; 2023) and reducing memory usage (Feng et al., 2024) for independent predictions. By contrast, our method targets efficiency in autoregressive joint sampling and evaluation – a largely unexplored area – and is complementary to these architectural improvements. More broadly, transformer-based probabilistic models increasingly frame Bayesian inference as in-context learning, performing tasks such as posterior approximation and predictive density estimation by conditioning on context observations (Mittal et al., 2023; Gloeckler et al., 2024; Reuter et al., 2025; Chang et al., 2025; Whittle et al., 2025; Mittal et al., 2025). The effectiveness of this paradigm has led to tabular foundation models such as TabPFN (Hollmann et al., 2023; 2025) and TabICL (Qu et al., 2025; 2026), which perform in-context learning over table rows using the same attention mechanisms as standard TNPs and PFNs. Our buffer integrates naturally with all these models.

**Autoregressive joint density estimation.** Autoregressive approaches are widely used for joint density estimation, from neural autoregressive density estimators (Larochelle & Murray, 2011; Uria et al., 2016; Germain et al., 2015) to normalizing flows (Kingma et al., 2016; Papamakarios et al., 2017; Huang et al., 2018; De Cao et al., 2020; Patacchiola et al., 2024) and order-agnostic autoregressive models (Uria et al., 2014; Hoogeboom et al., 2022; Liu et al., 2024). Most directly related is the Autoregressive Transformer NP (TNP-A; Nguyen & Grover, 2022), which duplicates targets into queries and observed values for both training and inference. We recognize that this duplication is only needed for predictive density evaluation, not training. Bruinsma et al. (2023) showed that deploying standard NPs autoregressively improves joint predictions but requires expensive context re-encoding at each step. Our buffer combines insights from both approaches: like TNP-A, we en-

able parallel predictive density evaluation, and like Bruinsma et al. (2023), we model autoregressive dependencies while training on independent targets, but our separate buffer architecture avoids both TNP-A's training overhead and the re-encoding bottleneck.

**Connection to other generative modeling techniques.** Modern generative models follow two main paradigms: diffusion and flow-matching models (Sohl-Dickstein et al., 2015; Ho et al., 2020; Song et al., 2021; Lipman et al., 2023) that generate samples through continuous-time dynamics, and autoregressive transformers (Radford et al., 2018; Brown et al., 2020) that generate sequences token-by-token with cached key-value states. Our buffer brings the caching efficiency of autoregressive transformers to NPs and PFNs: encode the set-based context once (analogous to a prompt) and generate efficiently through cached representations, rather than re-encoding the entire context at each autoregressive step. Recent work has shown that these paradigms can be combined (Tang et al., 2025; Arriola et al., 2025; Wu et al., 2025), suggesting future extensions. Recent work has also begun unifying masked diffusion (Lou et al., 2024; Shi et al., 2024; Sahoo et al., 2025) with any-order autoregressive models. Sahoo et al. (2026) exploit the connection between masked diffusion models and any-order autoregressive models to introduce KV caching for masked diffusion, enabling significant inference speedups while preserving parallel generation. In a related vein, Partition Generative Models (PGMs; Deschenaux et al., 2025) employ self-attention among observed tokens and cross-attention from unmasked to masked tokens – mirroring the context-target attention pattern in TNPs and PFNs, and pointing to a structural convergence between these model families.

## 5 EXPERIMENTS

We evaluate the buffer's efficiency gains and predictive fidelity across four domains: regression on synthetic functions, interpolation of real-world EEG data, Bayesian model selection on a multisensory perception model, and pre-training of a tabular foundation model. We first conduct wall-clock and memory benchmarks, then assess predictive performance across these varied domains.

**Baselines.** We compare against models spanning the efficiency-expressivity tradeoff, all configured with matched parameter counts, same input embeddings and output prediction heads unless noted otherwise (details in Appendix B). *TNP-D* (Nguyen & Grover, 2022) assumes conditional independence between targets; we evaluate it both with standard parallel decoding (*TNP-D-Ind*, fast but limited) and with autoregressive deployment (*TNP-D-AR*, expressive but requires sequential re-encoding). *TNP-ND* models target dependencies via a multivariate Gaussian, enabling one-pass joint predictive density evaluation but limiting expressivity. *TNP-A* uses causal self-attention for full autoregressive modeling but suffers from slow sequential sampling and high training cost. Additional task-specific baselines are introduced as needed. TNP-ND aside, all models use a Gaussian mixture model output head with 20 mixture components unless stated otherwise.

**Computational efficiency.** Our method trades exact set-based AR updates for efficiency. We benchmark wall-clock time for: **(i)** autoregressive sampling, **(ii)** joint predictive density evaluation, and **(iii)** a full training step (forward and backward pass), as well as **(iv)** peak memory usage. All measurements use a unified codebase and run on a single NVIDIA L40S GPU. We optimized all baselines beyond their public versions with KV caching, FlashAttention-2 (Dao, 2023), and compilation, achieving $3 - 10\times$ speedups over original implementations to ensure fair comparison. For our method, we developed a custom Triton kernel to optimize memory access during batched sampling (details in Appendix C). Benchmarks in Fig. 1 use model architectures matching subsequent experiments with buffer size $K = 16$. For sampling and predictive density evaluation: $M = 16$ targets, batch size $B = 256$. For training: $M = 256$ targets, batch size $B = 128$.

Our method achieves a superior efficiency profile compared to expressive baselines. *For autoregressive sampling* (Fig. 1, top left), our method is $3 - 20\times$ faster than the fully autoregressive TNP-A and TNP-D-AR. While TNP-D-Ind and TNP-ND are faster, they cannot capture complex predictive dependencies, as shown later in this section. *For predictive density evaluation* (Fig. 1, top center), our method's speed is on par with the highly parallel TNP-A and is a factor of $K\times$ faster than the sequential TNP-D-AR. *For training speed* (Fig. 1, top right), the overhead of our method is minimal, resulting in a training step time comparable to the fastest baselines (TNP-D, TNP-ND) and $4 - 12\times$ faster than TNP-A, which incurs a significant computational cost due to its architecture. *For memory*

Table 1: **Average predictive density (↑) results on synthetic functions and EEG example.** Mean and (SEM) over various functions and context sizes $N$, for $M = 16$ targets. See Appendix D.5 for evaluation details and Table A2 for results with larger $M$. TNP w/ buffer with $K$=1 tracks the best method; $K$=16 (fast) incurs only a slight drop in most cases.

| | TNP-D | | TNP-ND | TNP-A | TNP w/ buffer (ours) | |
| | AR | Ind | | | $K$=16 (fast) | $K$=1 (slow) |
|---|---|---|---|---|---|---|
| GP | 2.57 (0.020) | 2.22 (0.022) | 0.80 (0.082) | 2.24 (0.018) | 2.51 (0.019) | 2.56 (0.019) |
| Sawtooth | 1.05 (0.004) | 0.94 (0.005) | -0.43 (0.008) | 0.98 (0.004) | 1.00 (0.005) | 1.09 (0.004) |
| EEG-Int | 0.51 (0.013) | 0.36 (0.014) | 0.46 (0.011) | 0.58 (0.014) | 0.52 (0.013) | 0.54 (0.014) |
| EEG-For | 1.07 (0.004) | -0.74 (0.008) | -0.04 (0.005) | 1.23 (0.003) | 0.85 (0.004) | 1.21 (0.003) |

Table 2: **Multisensory causal inference model comparison and prediction results.** For *model selection*, we use two metrics: log marginal likelihood root mean-squared error (LML RMSE) against ground-truth, and difference in LML between $\rho = 4/3$ and $\rho = 1$, reported as RMSE ($\Delta$LML RMSE). For *data prediction*, we report average predictive density estimates (Average LL) for $M = 16$ targets, computed using the model selected by the model-selection task. See Appendix D.5 for additional details and evaluations.

| | TNP-D | | TNP-ND | TNP-A | TNP w/ buffer (ours) | |
| | AR | Ind | | | $K$=16 (fast) | $K$=1 (slow) |
|---|---|---|---|---|---|---|
| LML RMSE (↓) | 3.10 (0.005) | 86.96 (0.000) | 208.51 (0.041) | 4.75 (0.012) | 3.56 (0.004) | 3.47 (0.004) |
| $\Delta$LML RMSE (↓) | 2.44 (0.008) | 36.18 (0.000) | 25.60 (0.023) | 3.29 (0.019) | 2.60 (0.010) | 2.59 (0.011) |
| Average LL (↑) | -2.76 (0.024) | -2.77 (0.025) | -3.12 (0.016) | -2.76 (0.024) | -2.76 (0.024) | -2.76 (0.024) |

*usage* (Fig. 1, bottom left), our method requires $6 - 7\times$ less VRAM than TNP-D-AR and TNP-A at large context sizes ($N = 1024$), scaling efficiently due to only needing to cache a single context independent of batch size. *For predictive performance* (Fig. 1, bottom center), we show normalized scores[4] averaged across the six tasks presented in Tables 1 and 2; our method closely matches the expressive autoregressive baselines (TNP-D-AR, TNP-A) while substantially outperforming TNP-D-Ind and TNP-ND. We provide additional results, including benchmarks across a wider range of batch and target sizes and memory usage comparison, in Appendix C.

**Synthetic functions.** We consider two prediction tasks: **(i)** functions drawn from Gaussian processes (GPs; Rasmussen & Williams, 2006) where the *kernel type* is sampled from a set, and its hyperparameters, and **(ii)** a non-Gaussian sawtooth process with discontinuous derivatives. All models are trained and evaluated on distinct draws from these processes (see Appendix D.2). *Results:* As shown in Table 1, TNP w/ buffer ($K = 16$)[5] achieves performance comparable to TNP-D-AR while providing substantial speedups (Fig. 1), indicating that the buffer captures relevant target dependencies even without full context re-encoding. To verify that buffer training does not degrade AR capability, we deploy our model with $K = 1$ (empty buffer), matching the performance of TNP-D-AR exactly.

**Electroencephalogram (EEG) data.** Following Markou et al. (2022) and Bruinsma et al. (2023), we train TNPs on EEG time series data (Zhang et al., 1995). Each trial contains 256 regularly sampled measurements across 7 correlated channels. See Appendix D.2 for dataset details. We train on an interpolation setting and evaluate on both forecasting and interpolation tasks. Interpolation uses random context/target splits; forecasting uses the first $N$ points as context and the next $M$ as targets (Appendices D.2 and D.5). As shown in Table 1, our method with $K$=16 is comparable to TNP-D-AR for interpolation but shows a larger gap for forecasting; deploying with $K$=1 (no buffer) closes this gap. In both settings, our method is substantially better than TNP-D-Ind and TNP-ND. Additional results (larger $M$; permutation effects in forecasting) are in Appendices E.2 and E.3.

---

[4]We compute normalized scores for each task by linearly rescaling the average log predictive densities so that the worst-performing method scores 0 and the best-performing method scores 1.

[5]See Appendix H.5 for ablations on different buffer sizes.

**Multisensory causal inference model comparison and data prediction.** We evaluate our method on a popular computational neuroscience model that determines how the brain combines sensory stimuli from different sources (Körding et al., 2007). Using publicly available data from an audio-visual localization experiment (Liu et al., 2025), we consider two model variants, each with 7 free parameters, differing only in their auditory recalibration parameter $\rho \in \{1, 4/3\}$, and evaluate two tasks: **(1) Model selection.** For each method, we train two models on simulators with $\rho = 1$ and $\rho = 4/3$. We use the trained models for the challenging task of computing the log marginal likelihood (LML) of real experimental data, which requires evaluating the joint likelihood (Murphy, 2012):

$$\text{LML} = \log p(y_{1:N}|\mathbf{x}_{1:N}) = \sum_{i=1}^{N} \log p(y_i|\mathbf{x}_i, \{(\mathbf{x}_j, y_j)\}_{j<i}) \tag{6}$$

which is inherently an autoregressive prediction task, as each prediction conditions on all previous data points, so it is perfectly suited for our models. For each dataset, we estimate the ground-truth LML for both $\rho = 1$ and $\rho = 4/3$ using S-VBMC, a method proven effective on similar problems (Acerbi et al., 2018; Silvestrin et al., 2025). We report LML RMSE and $\Delta$LML RMSE (the *difference* between model metrics, useful for model comparison) in Table 2. **(2) Data prediction.** Using the model selected in (1), we predict outputs on the real dataset and report average log-predictive densities (Table 2). See Appendix D.3 for experimental details and Appendix D.5 for evaluation settings.

*Results.* We evaluate our method using data from the 15 participants of the original study, extracting two non-overlapping subsets of 400 experimental trials each (400 data points), resulting in a total of 30 datasets. The model trained with $\rho = 4/3$ generally achieves better (higher) LML than $\rho = 1$, aligning with the original finding that participants are remapping their auditory space to match the visual range (Liu et al., 2025). Fig. 3 shows that the LML and $\Delta$LML approximations obtained with our method are remarkably close to the ground-truth. Furthermore, our method performs on par with TNP-D-AR and outperforms all other baselines on model comparison (Table 2). All models except TNP-ND perform similarly on the data prediction task. For additional results, see Appendix F.

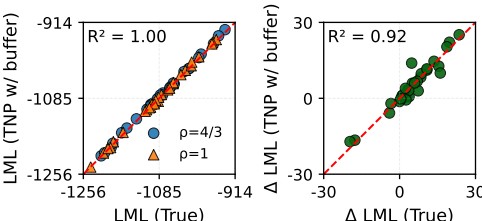

Figure 3: Multisensory causal inference model comparison versus ground-truth. (Left) Log marginal likelihood (LML) comparison for both $\rho = 1$ and $\rho = 4/3$. (Right) LML difference ($\rho = 4/3 - \rho = 1$) comparison. Our method closely aligns with the ground-truth.

**Small-scale tabular foundation model.** We integrate our autoregressive buffer into the TabICL foundation model architecture (Qu et al., 2025). While the original work focused on classification, we pre-train our model from scratch for regression tasks. We reuse TabICL's set encoder to compute feature embeddings upfront and focus modifications on the final *dataset-wise in-context learning transformer*. Our core methodological contribution is the buffer mechanism, implemented by a structured attention mask. This allows the model to condition on its recent predictions by storing them in a dynamic buffer. We pre-train this architecture on synthetic data from a *structural causal model* (SCM) prior (Hollmann et al., 2023; Qu et al., 2025), where each training instance is formed by partitioning datasets into distinct sets of context, buffer, and target points. Our network size and training scale are comparable to the original TabPFN (Hollmann et al., 2023); the model is pre-trained on 10.24 million synthetic datasets containing 1 to 10 features and 8 to 1024 context points, with a buffer size of $K = 32$. Full details are provided in Appendix D.4.

*Results.* We evaluate on six UCI and Kaggle datasets:[6] Individual Household Electric Power Consumption, Gas Turbine CO and NOx Emission, Bike Sharing, Jena Climate, Power Consumption of Tetouan City, and California Housing Prices. We form 16 random context/target splits per dataset for both $N = 16$ and $N = 1024$ context sizes and $M = 32$ targets under interpolation (Int) and forecasting (For) tasks; the latter for time-series datasets only (all excluding California Housing Prices). We compare three inference modes from the same pre-trained model: *"Ind"* (independent predictions),

---

[6]UCI: https://archive.ics.uci.edu/. Kaggle: https://www.kaggle.com/.

Table 3: **Average log-predictive density (↑) results on UCI datasets with TabICL.** We evaluate our AR buffer integrated into a TabICL foundation model against independent and standard AR baselines. Performance is measured on interpolation (Int) and forecasting (For) tasks across six real-world datasets. Results are reported as mean and standard error over 16 randomly sampled mini-datasets ($M = 32$) in low context ($N = 16$) and high context ($N = 1024$) settings.

| | **LOW CONTEXT REGIME** ($N = 16$) | | | | | | | | | | |
| | Electric Cons. | | Gas Turbine | | Bike Sharing | | Tetouan | | Jena | | Cali. |
| | Int | For | Int | For | Int | For | Int | For | Int | For | Int |
|---|---|---|---|---|---|---|---|---|---|---|---|
| Independent | 0.38 (0.22) | -2.87 (0.77) | -0.65 (0.15) | -1.46 (0.26) | 0.84 (0.11) | -0.04 (0.17) | -0.59 (0.14) | -4.71 (0.46) | 0.10 (0.12) | -4.53 (1.07) | -1.31 (0.07) |
| Standard AR | 0.88 (0.19) | -0.97 (0.52) | -0.48 (0.13) | -1.00 (0.18) | 1.43 (0.11) | 1.02 (0.13) | -0.09 (0.14) | -2.39 (0.22) | 0.64 (0.12) | -2.15 (0.35) | -1.17 (0.08) |
| AR w/ buffer | 0.78 (0.19) | -1.00 (0.51) | -0.48 (0.13) | -0.98 (0.17) | 1.31 (0.10) | 0.94 (0.13) | -0.10 (0.15) | -2.41 (0.23) | 0.55 (0.12) | -2.15 (0.34) | -1.16 (0.09) |

| | **HIGH CONTEXT REGIME** ($N = 1024$) | | | | | | | | | | |
| | Electric Cons. | | Gas Turbine | | Bike Sharing | | Tetouan | | Jena | | Cali. |
| | Int | For | Int | For | Int | For | Int | For | Int | For | Int |
|---|---|---|---|---|---|---|---|---|---|---|---|
| Independent | 1.78 (0.06) | 1.64 (0.18) | -0.01 (0.16) | -0.60 (0.29) | 2.54 (0.05) | 2.32 (0.07) | 0.36 (0.07) | -1.12 (0.35) | 2.01 (0.06) | 1.56 (0.19) | -0.44 (0.08) |
| Standard AR | 1.78 (0.06) | 1.70 (0.18) | -0.01 (0.16) | -0.47 (0.27) | 2.54 (0.05) | 2.40 (0.10) | 0.36 (0.07) | -0.08 (0.22) | 2.01 (0.06) | 1.80 (0.13) | -0.44 (0.08) |
| AR w/ buffer | 1.79 (0.06) | 1.70 (0.18) | -0.01 (0.16) | -0.48 (0.27) | 2.53 (0.05) | 2.39 (0.06) | 0.36 (0.06) | -0.12 (0.23) | 2.01 (0.06) | 1.64 (0.16) | -0.44 (0.08) |

*"Standard AR"* (step-by-step autoregression, $K=1$ equivalent), and *"AR w/ buffer"* (*ours*, $K=32$). Results in Table 3 show that both AR modes consistently outperform independent predictions in low context settings and in high context forecasting. Crucially, AR w/ buffer matches standard AR within standard errors, demonstrating that the buffer preserves dependencies while enabling efficient autoregressive inference. In Appendix G, we provide results for the $N = 256$ setting.

## 6 DISCUSSION

We introduce a causal autoregressive buffer that decouples one-time context encoding from lightweight sequential updates in transformer-based probabilistic models. By caching context keys/values and routing target-to-target dependencies through a causal buffer, we reduce the attention cost from $\mathcal{O}(K(N+K)^2)$ to $\mathcal{O}(N^2 + NK + K^2)$. Across synthetic functions, EEG interpolation, multisensory modeling, and tabular prediction, our method matches autoregressive baselines while achieving up to $20\times$ faster joint sampling with minimal additional training cost over standard models and up to $10\times$ lower training cost than autoregressive-specific baselines.

There are several limitations. First, cost grows with $K$. Runtime and memory include an $\mathcal{O}(K^2)$ term from causal self-attention in the buffer, and we currently learn a fixed set of buffer positional embeddings. Scaling to longer horizons without growing training complexity may be possible via rotary position embeddings (RoPE; Su et al., 2024) or attention biasing (ALiBi; Press et al., 2022). Second, for long buffers, quality can drift relative to exact AR that re-encodes the context at each step. Exploring the draft-verify process from speculative decoding (Leviathan et al., 2023; Chen et al., 2023) could enable adaptive inference strategies using the buffer for improved performance.

A practical strength of our method is its plug-and-play applicability: the buffer is implemented via attention masks and token roles, as demonstrated by its integration into the TabICL architecture. While we currently perform joint training of the base model and buffer, our method could be directly applied to pretrained NPs/PFNs. Parameter-efficient fine-tuning (Houlsby et al., 2019; Hu et al., 2022) could offer a direct path to enable buffered inference without full retraining. We also leave to future work a deeper exploration of alternative attention backbones (e.g., Jaegle et al., 2021; see Appendix H.4) and broader inference tasks (Chang et al., 2025).

ACKNOWLEDGEMENT

This work was supported by the Research Council of Finland (Flagship programme: Finnish Center for Artificial Intelligence, FCAI) and ELISE Networks of Excellence Centres (EU Horizon:2020 grant agreement 951847). The project is also supported by the EuroHPC Joint Undertaking and its members including top-up funding by Ministry of Education and Culture. CH and SK were supported by Business Finland (VirtualLab4Pharma, grant agreement 3597/31/2023) and the European Union (Horizon Europe, grant agreement 101214398, ELLIOT). NL was funded by Business Finland (project 3576/31/2023) and LUMI AI Factory (EU Horizon Europe Joint Undertaking and its members including top-up funding by Ministry of Education and Culture). YY was supported by the Ministry of Education and Culture's Doctoral Education Pilot under Decision No. VN/3137/2024-OKM-6 (The Finnish Doctoral Program Network in Artificial Intelligence, AI-DOC). SK was supported by UKRI Turing AI World-Leading Researcher Fellowship (EP/W002973/1). LA was supported by Research Council of Finland grants 356498 and 358980, the latter also supporting FS. The authors also acknowledge the research environment provided by ELLIS Institute Finland.

We acknowledge Verda for providing computational resources. Additionally, we acknowledge CSC – IT Center for Science, Finland, for computational resources provided by the LUMI supercomputer, owned by the EuroHPC Joint Undertaking and hosted by CSC and the LUMI consortium (LUMI projects 462000943 and 462000874). Access was provided through the Finnish LUMI-OKM allocation. We also acknowledge the computational resources provided by the Aalto Science-IT Project from Computer Science IT.

Funded by the European Union. Views and opinions expressed are however those of the author(s) only and do not necessarily reflect those of the European Union or the granting authority. Neither the European Union nor the granting authority can be held responsible for them.

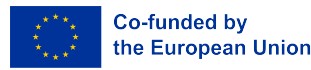 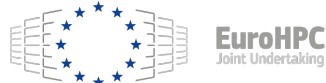

ETHICS STATEMENT

This work uses only publicly available datasets and synthetic simulators, with no sensitive data involved. The methods are for research purposes and pose no foreseeable ethical risks. We have followed the ICLR Code of Ethics.

REPRODUCIBILITY STATEMENT

We will release the code, including the training and evaluation pipelines, as well as configuration files, in the repository linked below.[7] All experiments use public datasets or, when applicable, a simulator for synthetic data. Algorithmic details are presented in Algorithms 1 and 2, and all hyperparameters and training schedules are specified in the configuration files and documented in the appendix. Ablation studies are also reported in the appendix. We do not release pretrained weights, and no special data licenses or usage constraints apply.

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

# Table of Contents

# A  METHOD DETAILS

This appendix spells out the modules used in Eq. (3), the single block-sparse attention mask that implements requirements (R1)–(R4), and the exact procedures for autoregressive sampling and one-pass joint log-likelihood evaluation.

## A.1  MODULES AND NOTATION

Our method uses three sets of tokens: context $\mathcal{C}$, buffer $\mathcal{B}$, and targets $\mathcal{T}$, of sizes $N, K, M$, respectively. Throughout this paper, let

$$\mathbf{E}_x : \mathcal{X} \to \mathbb{R}^d, \quad \mathbf{E}_y : \mathcal{Y} \to \mathbb{R}^d, \quad \mathbf{a} : \{1, \ldots, K\} \to \mathbb{R}^d$$

denote learned embeddings for inputs, outputs, and buffer positions. In addition, we introduce role embeddings that indicate token type, denoted by $e_{\text{ctx}}^{\text{role}}$, $e_{\text{buf}}^{\text{role}}$, and $e_{\text{tgt}}^{\text{role}}$ for context, buffer, and target tokens, respectively.

**Context encoder $\mathbf{r}_{\mathcal{C}}$.** Given context pairs $\mathcal{C} = \{(\mathbf{x}_n, y_n)\}_{n=1}^N$, construct context tokens: $e_n^{\text{ctx}} = \mathbf{E}_x(\mathbf{x}_n) + \mathbf{E}_y(y_n) + e_{\text{ctx}}^{\text{role}}$, process them with bidirectional MHSA (no positional embeddings), and cache per-layer keys/values:

$$\{\text{KV}_{\mathcal{C}}^{\ell}\}_{\ell=1}^L \; = \; \mathbf{r}_{\mathcal{C}}(\mathcal{C}) \quad \text{(computed once; immutable)}.$$

**Buffer encoder $\mathbf{r}_{\mathcal{B}}$.** For a buffer prefix $\mathcal{B}_{1:k} = \{(\mathbf{x}_j^{\star}, y_j^{\star})\}_{j=1}^k$, form tokens $e_j^{\text{buf}} = \mathbf{E}_x(\mathbf{x}_j^{\star}) + \mathbf{E}_y(y_j^{\star}) + \mathbf{a}(j) + e_{\text{buf}}^{\text{role}}$, then apply *strictly causal* MHSA on $\{e_j^{\text{buf}}\}_{j \leq k}$ so that each token is restricted to attend only to earlier tokens in the sequence, and in addition, each token performs cross-attention to the cached context $\{\text{KV}_{\mathcal{C}}^{\ell}\}$. This yields per-layer $\text{KV}_{\mathcal{B}_{1:k}}$ that we update incrementally at inference:

$$\{\text{KV}_{\mathcal{B}_{1:k}}^{\ell}\}_{\ell=1}^L \; = \; \mathbf{r}_{\mathcal{B}}\left(\mathcal{B}_{1:k}, \mathbf{r}_{\mathcal{C}}(\mathcal{C})\right).$$

**Target decoder $\mathbf{r}_{\text{tgt}}$ and prediction head.** For a target input $\mathbf{x}_m^{\star}$ we create a query token $e_m^{\text{tgt}} = \mathbf{E}_x(\mathbf{x}_m^{\star}) + e_{\text{tgt}}^{\text{role}}$. The target decoder $\mathbf{r}_{\text{tgt}}$ performs a *single cross-attention* from $e_m^{\text{tgt}}$ to the *concatenated* keys/values of the context cache $\{\text{KV}_{\mathcal{C}}^{\ell}\}$ and the *visible* buffer prefix $\{\text{KV}_{\mathcal{B}_{1:v_m}}^{\ell}\}$, followed by normalization and an MLP:

$$\mathbf{h}_m = \mathbf{r}_{\text{tgt}}\left(e_m^{\text{tgt}}, \left[\{\text{KV}_{\mathcal{C}}^{\ell}\}, \{\text{KV}_{\mathcal{B}_{1:v_m}}^{\ell}\}\right]\right), \qquad \phi_m = \psi(\mathbf{h}_m),$$

where $\psi$ is the distribution head (e.g., the mixture-of-Gaussian head).

## A.2  TRAINING MASK THAT IMPLEMENTS (R1)–(R4)

We concatenate tokens as $[\mathcal{C}, \mathcal{B}, \mathcal{T}]$ with sizes $N, K$, and $M$, respectively, and use one block-sparse attention mask consisting of the following *five* unmasked sections (everything else is masked):

**(1) Self-attention within context.** Context tokens attend bidirectionally to other context tokens. Context never attends to buffer or targets (context is read-only outside this block).

**(2) Buffer reads context (cross-attention).** Each buffer token can read (attend to) all context tokens. This lets the buffer incorporate task information from the cached context while keeping the context cache immutable.

**(3) Causal self-attention within the buffer.** Within the buffer itself, attention is strictly causal: a buffer token at position $j$ can only read earlier buffer positions $< j$ (no future reads). This encodes the autoregressive dependency among realized targets.

**(4) Targets read context (cross-attention).** Each target query can read the entire cached context. There are no edges between targets.

**(5) Targets read buffer (prefix only, cross-attention).** Each target query can read only a *visible prefix* of the buffer. The visible prefix length for target $m$ is $v_m$: *training (teacher forcing):* we set $v_m = 0$ for 50% of targets and sample $v_m \sim \text{Uniform}\{1, \ldots, K\}$ for the rest (the curriculum);

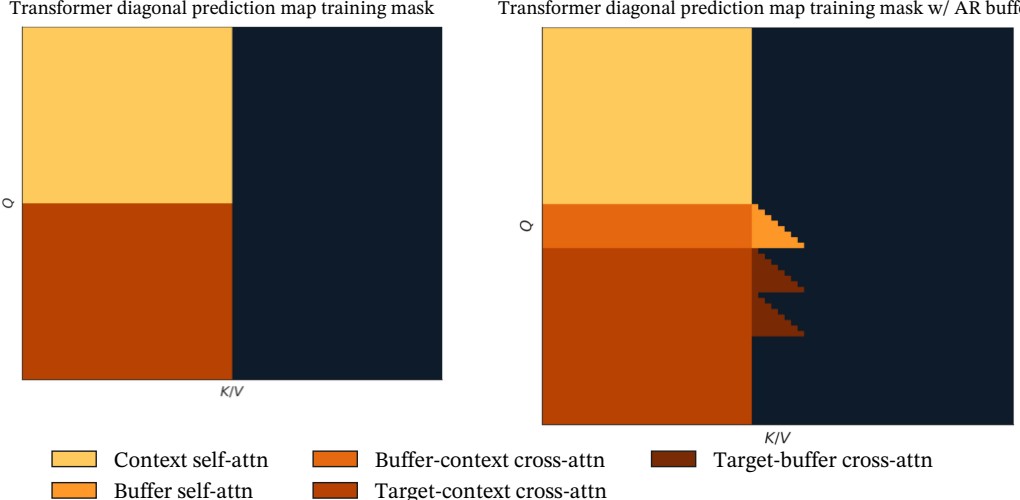

Figure A1: **Block-sparse attention masks with and without an autoregressive buffer.** *Left:* a diagonal prediction-map transformer (e.g., TNP/PFN): the context attends to itself and each target reads the entire context. *Right:* our buffered variant inserts an autoregressive memory $\mathcal{B}$ between context and targets, adding three blocks: (i) buffer reads context (ii) *causal* self-attention within buffer (iii) target reads varying number of elements from start of buffer, depending on curriculum.

---

**Algorithm 1** Autoregressive sample generation for $K$ targets

---

**Require:** Context $\mathcal{C} = \{(x_n, y_n)\}_{n=1}^N$, target inputs $\{x_k^\star\}_{k=1}^K$
1: $\{\text{KV}_\mathcal{C}^\ell\} \leftarrow \mathbf{r}_\mathcal{C}(\mathcal{C})$                                           $\triangleright \mathcal{O}(N^2)$; cached
2: Initialize $\{\text{KV}_{\mathcal{B}_{1:0}}^\ell\}$                                      $\triangleright$ empty buffer cache
3: **for** $k = 1$ to $K$ **do**
4:      $\mathbf{h}_k \leftarrow \mathbf{r}_{\text{tgt}}\left(\mathbf{E}_x(x_k^\star) + e_{\text{tgt}}^{\text{role}}, \left[\{\text{KV}_\mathcal{C}^\ell\}, \{\text{KV}_{\mathcal{B}_{1:k-1}}^\ell\}\right]\right)$
5:      Sample $y_k^\star \sim p_\theta(\cdot; \psi(\mathbf{h}_k))$
6:      Append $(x_k^\star, y_k^\star)$; update $\{\text{KV}_{\mathcal{B}_{1:k}}^\ell\}$ (strictly causal)
7: **end for**
8: **return** $\{y_k^\star\}_{k=1}^K$

---

*sampling:* at step $k$, the active query sees the realized prefix $k{-}1$; *one-pass joint log-likelihood:* packed queries use $v_m = m{-}1$ to recover the autoregressive chain in a single forward pass.

All other connections are masked: context never reads buffer or targets; targets never read targets; and buffer never reads targets. This single pattern implements the four requirements from the main text—immutable context, strictly causal buffer, unidirectional flow out of context, and target access to (context + visible buffer). See Fig. A1 for the diagram.

**Complexity.** Under this mask, a full prediction pass costs $\mathcal{O}(N^2 + NK + K^2)$ attention operations per layer: one-time $\mathcal{O}(N^2)$ for $\mathcal{C}$, $\mathcal{O}(NK)$ for reads from $\mathcal{C}$, and $\mathcal{O}(K^2)$ for causal buffer self-attention. This replaces the $\mathcal{O}(K(N+K)^2)$ cost of naive AR re-encoding. Packing $B$ target orders in parallel (for order averaging) isolates the $B$ buffer sets while sharing the context cache, yielding $\mathcal{O}(N^2 + B(NK + K^2))$.

A.3    ALGORITHMS FOR AUTOREGRESSIVE SAMPLING AND LOG-LIKELIHOOD EVALUATION

We include here the pseudocode for the main procedures used in our method. Algorithm 1 details the autoregressive sampling procedure, and Algorithm 2 presents the joint likelihood evaluation.

---

**Algorithm 2** Joint log-likelihood evaluation for $K$ targets

---

**Require:** Context $\mathcal{C} = \{(x_n, y_n)\}_{n=1}^{N}$, ordered targets $\{(x_k^\star, y_k^\star)\}_{k=1}^{K}$
1: $\{\mathrm{KV}_{\mathcal{C}}^{\ell}\} \leftarrow \mathbf{r}_{\mathcal{C}}(\mathcal{C})$                                                            $\triangleright \mathcal{O}(N^2)$; cached
2: Build all $K$ buffer tokens; compute $\{\mathrm{KV}_{\mathcal{B}_{1:K}}^{\ell}\}$ under causal mask
3: Build target queries $\{\mathbf{E}_x(x_k^\star) + e_{\mathrm{tgt}}^{\mathrm{role}}\}_{k=1}^{K}$
4: Mask: target $k$ sees $\mathcal{B}_{1:k-1}$ and all of $\mathcal{C}$
5: Compute $\{\log p_k\}_{k=1}^{K}$;
6: **return** $\sum_{k=1}^{K} \log p_k$

---

## B   Transformer Neural Process Baselines Details

We summarize the baseline transformer neural process (TNP) variants used in our comparisons, following Nguyen & Grover (2022). Architectural hyperparameters appear in Appendix D.1.

### B.1   TNP-D

This model takes as input a context set $\{(\mathbf{x}_n, y_n)\}_{n=1}^{N}$ and a target set $\{\mathbf{x}_m^\star\}_{m=1}^{M}$. Similar to Appendix A, the context embeddings $e_n^{\mathrm{ctx}}$ are processed with bidirectional MHSA with no positional encodings. Each target is decoded by:

$$\mathbf{h}_m = \mathbf{r}_{\mathrm{tgt}}\Big(e_m^{\mathrm{tgt}}, \ \mathbf{r}_{\mathcal{C}}(\mathcal{C})\Big), \qquad \phi_m = \psi(\mathbf{h}_m),$$

where $\psi$ is the distribution head (Gaussian as in the original paper; we primarily use a mixture of Gaussians). The left panel of Fig. A1 shows the training mask for TNP-D. This model is trained via maximum likelihood estimation of independent targets given a fixed context set.

At deployment, the decoding can be independent or autoregressive, yielding TNP-D-Ind and TNP-D-AR methods. TNP-D-Ind decodes all targets independently in a single pass. It is fast (context and targets encoded once), but cannot capture dependencies between targets.

TNP-D-AR decodes targets sequentially, appending each sampled $(\mathbf{x}_m^\star, y_m^\star)$ to the context. This captures joint structure but requires re-encoding the growing set at each step. TNP-D-Ind is invariant to target order; TNP-D-AR is order-sensitive, so we approximate the predictive distribution by averaging over multiple target orderings.

### B.2   TNP-ND

This model encodes the context set once and decodes all targets simultaneously by parameterizing a joint multivariate Gaussian distribution over the outputs. The embedder and transformer backbone are identical to those of TNP-D-Ind:

$$\mathbf{h}_m = \mathbf{r}_{\mathrm{tgt}}\Big(e_m^{\mathrm{tgt}}, \ \mathbf{r}_{\mathcal{C}}(\mathcal{C})\Big).$$

Then the joint distribution is obtained via

$$\phi = \psi_{ND}(\mathbf{h}_1, \ldots, \mathbf{h}_M),$$

where $\psi_{ND}$ is the multivariate Gaussian head that outputs both a mean vector and valid covariance matrix. The mean is produced per target, and a lightweight self-attention head over the set of targets yields fixed-width embeddings that are transformed into a valid covariance factor. This design supports a variable number of targets and is invariant to target order.

The training optimizes the joint multivariate Gaussian likelihood of the target points. At inference, the joint samples and log-likelihood are computed in a single pass. This model is invariant to the order of target points.

### B.3   TNP-A

The key difference between this model and TNP-D is the attention mechanism on the target set. This model processes three sets: the context $\{(\mathbf{x}_n, y_n)\}_{n=1}^{N}$, the target $\{\mathbf{x}_m^\star\}_{m=1}^{M}$, and the observed

target $\{(\mathbf{x}_m^\star, y_m^\star)\}_{m=1}^M$. To differentiate, we denote the embeddings of $\{(\mathbf{x}_m^\star, y_m^\star)\}_{m=1}^M$ by $\{e_m^{y,\mathrm{tgt}}\}$. Similar to TNP-D, the context embeddings attend to each other. For the target set, each $e_m^{\mathrm{tgt}}$ attends to the context and the previous observed target embeddings $e_{j<m}^{y,\mathrm{tgt}}$. Likewise, the observed target embeddings attend to context and previous target embeddings (Fig. 2 of Nguyen & Grover 2022).

The target causal mask allows TNP-A to model the joint likelihood simultaneously in one single pass, assuming the observations are given (e.g., for training and test log-likelihood evaluations). For prediction generation, however, each sampled target needs to be re-encoded and attended for the generation of next targets, yielding a sequential re-encoding procedure. The causal mask on the target set is sensitive to the target order, and thus the final likelihood is an average over multiple random permutations. Note that this model processes duplicated target set–$\{\mathbf{x}_m^\star\}_{m=1}^M$ and an observed sequence $\{(\mathbf{x}_m^\star, y_m^\star)\}_{m=1}^M$; this creates significant computational overhead in both the training and the inference, particularly when the target set is large (see e.g. Appendix C and Figs. A7 to A9).

Compared to our method, TNP-A can be viewed as TNP-D with a '*frozen buffer*' $\{(\mathbf{x}_m^\star, y_m^\star)\}_{m=1}^M$ of size $K = M$ containing the observed targets. For likelihood evaluation where all sets are processed in one shot, the behavior of TNP-A and our approach are analogous, with the set $\{(\mathbf{x}_m^\star, y_m^\star)\}_{m=1}^M$ serving a role similar to our buffer. However, for AR sampling, TNP-A repeatedly re-encodes the full context and target sets after each sampled $y_m^\star$, whereas our method dynamically adapts to new samples. Moreover, since TNP-A does not afford a dynamic-size target set decoupled from the 'in-context' targets, training is much more expensive than our method (see Fig. 1 in the main text).

## C  COMPUTATIONAL EFFICIENCY DETAILS

This section provides additional empirical results to support the efficiency claims made in the main paper. We present an analysis of performance scaling with batch size, an ablation study of our custom kernel, a comparison against unoptimized open-source baselines, and further ablations on training time. In all subsequent plots, the absence of a data point for a given method indicates that the experiment failed due to an out-of-memory (OOM) error for that specific configuration.

### C.1  SCALING WITH BATCH SIZE

To analyze the effect of batch size $B$, we provide expanded results for autoregressive sampling and joint log-likelihood evaluation in Fig. A2 and Fig. A3, respectively. These plots show the wall-clock time as a function of the number of context points $N$ for various batch sizes. The results confirm that our method's performance advantage over autoregressive baselines like TNP-A is consistent and often widens as the context and batch size increase.

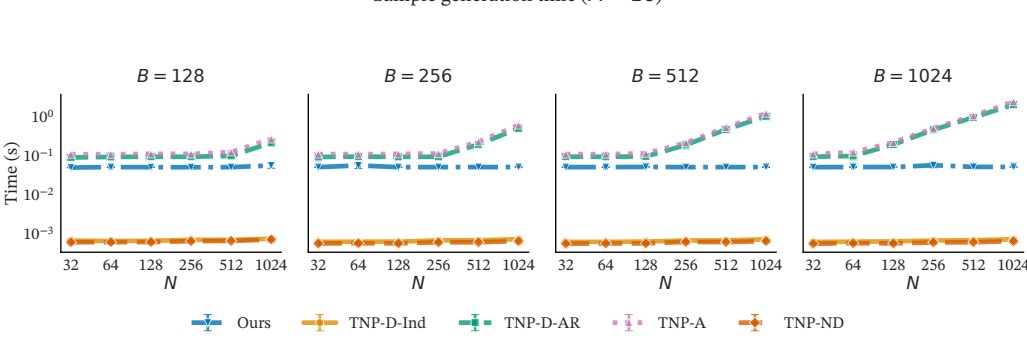

Figure A2: Autoregressive sampling time (log scale) versus context size $N$ for an expanded range of batch sizes $B$.

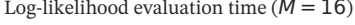

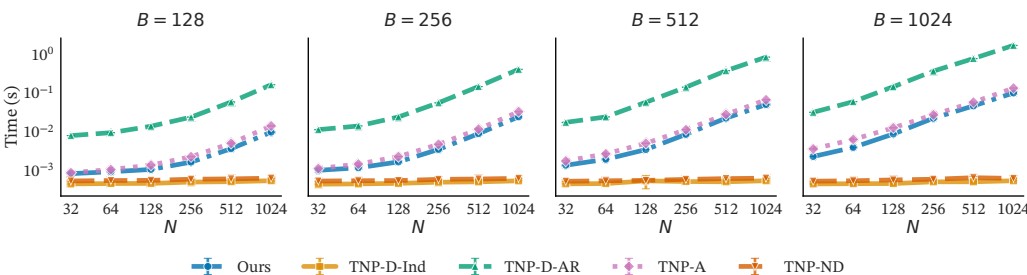

Figure A3: Joint log-likelihood evaluation time (log scale) versus context size $N$ for an expanded range of batch sizes $B$.

## C.2 IMPACT OF CUSTOM TRITON KERNEL

To isolate the contribution of our custom attention kernel, we compare the sampling time of our method with and without this optimization. The kernel is designed to accelerate a key computational step: the cross-attention between the batched target embeddings (batch size $B$) and the concatenation of a batched buffer cache with a *shared* context cache (batch size 1). A naive implementation would explicitly expand the context cache tensor $B$ times to match the batch dimension before the attention operation. This "expand" operation is memory-bandwidth intensive and creates a large, redundant intermediate tensor.

Our custom Triton kernel avoids this bottleneck by fusing the expansion and attention computations. The kernel loads the single context cache into fast SRAM and reuses it for each item in the batch, calculating the attention on-the-fly without ever materializing the full expanded tensor in slower global memory. As shown in Fig. A4, this memory-centric optimization provides a substantial speedup that grows with the batch size $B$.

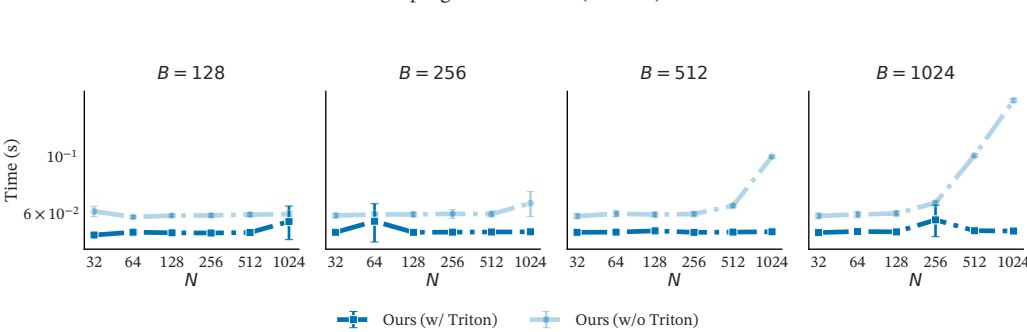

Figure A4: Ablation study for autoregressive sampling, comparing our method with and without the custom Triton kernel across different context and batch sizes.

## C.3 COMPARISON TO OPEN-SOURCE BASELINES

To demonstrate the fairness of our primary comparisons, we benchmark our optimized baseline implementations against their standard, publicly available versions. The results for sampling and likelihood evaluation are shown in Fig. A5 and Fig. A6. Our optimized baselines are consistently $3-10\times$ faster than their standard counterparts. This confirms that our method's performance gains are due to fundamental algorithmic advantages, not an unfair comparison against unoptimized code.

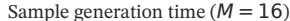

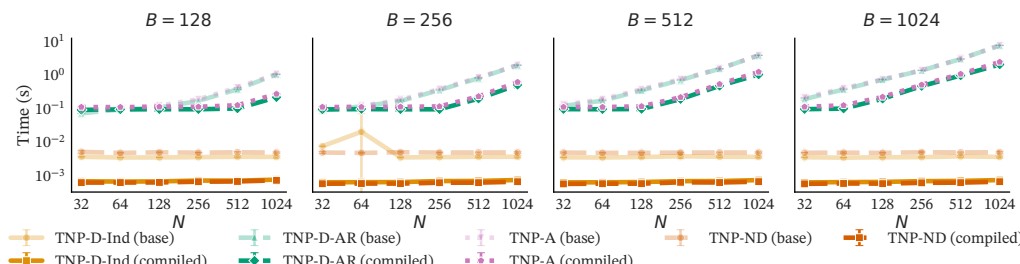

Figure A5: Comparison of our optimized baseline implementations against standard open-source versions for autoregressive sampling.

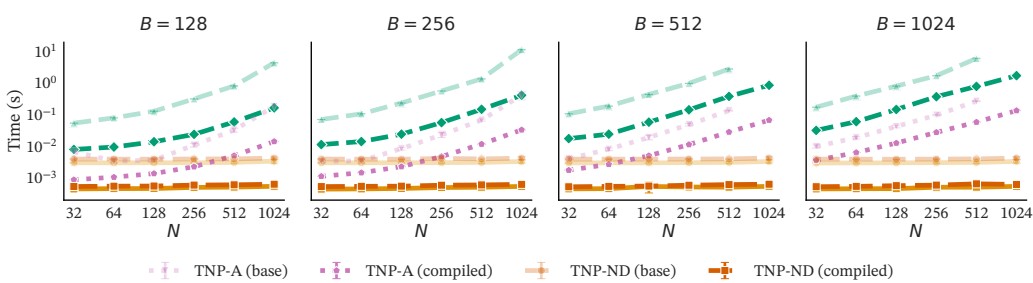

Figure A6: Comparison of our optimized baseline implementations against standard open-source versions for joint log-likelihood evaluation.

## C.4 TRAINING TIME SCALING

We further analyze the scaling of training step time with respect to the number of target points $M$ for different batch sizes. Figs. A7 to A9 show this relationship for batch sizes of 64, 128, and 256, respectively. The results show that as the context, target, or batch size increases, TNP-A becomes increasingly expensive to train relative to all other methods.

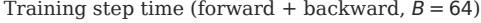

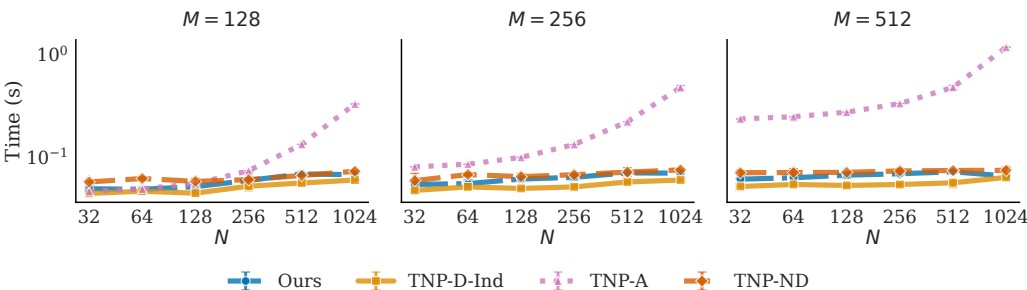

Figure A7: Training step time vs. number of target points $M$ for batch size $B = 64$.

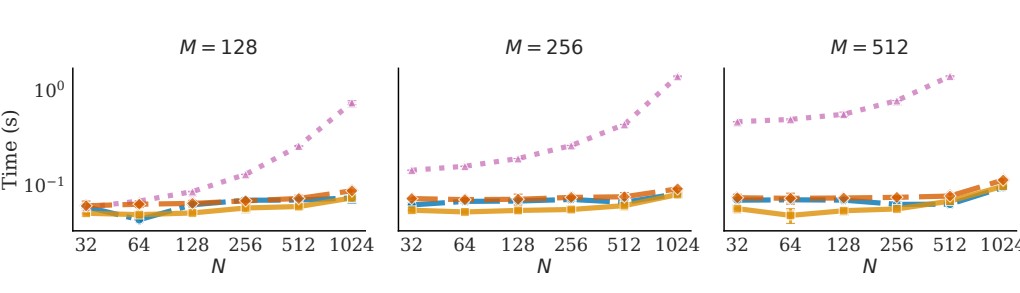

Figure A8: Training step time vs. number of target points $M$ for batch size $B = 128$.

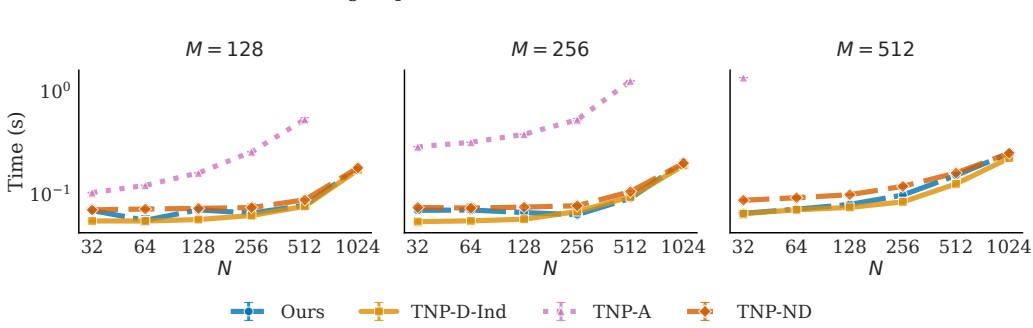

Figure A9: Training step time vs. number of target points $M$ for batch size $B = 256$.

## C.5 IMPACT OF ATTENTION PATTERNS ON TRAINING SPEED

A key difference between the baseline models is their compatibility with modern, efficient attention implementations. The causal attention mask required by TNP-A during training is incompatible with kernels like FlashAttention, forcing it to use PyTorch's standard, but slower, "math" attention backend. In contrast, models like TNP-D and ours can leverage these faster kernels.

In Appendix B, we discussed the duplicated processing of TNP-A on the target set, which incurs significant computational overhead. To determine if TNP-A's slow training is fundamental to its architecture or merely an artifact of this kernel incompatibility, we conduct a controlled ablation. We disable FlashAttention for *all* methods, forcing a fair comparison on the same standard PyTorch attention backend. The results, shown in Figs. A10 to A12, are unequivocal. Even on a level playing field, TNP-A's training time is orders of magnitude slower than all other methods. This confirms that its high computational cost is an inherent consequence of its autoregressive design, not just an implementation detail.

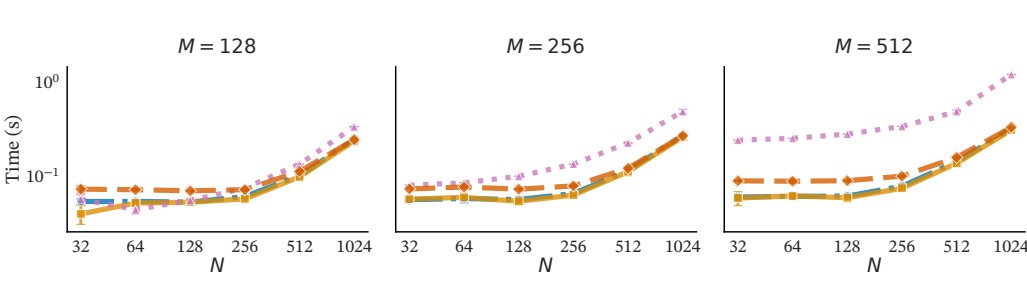

Figure A10: Training step time vs. number of target points $M$ using the standard PyTorch attention backend (FlashAttention disabled). Batch size $B = 64$.

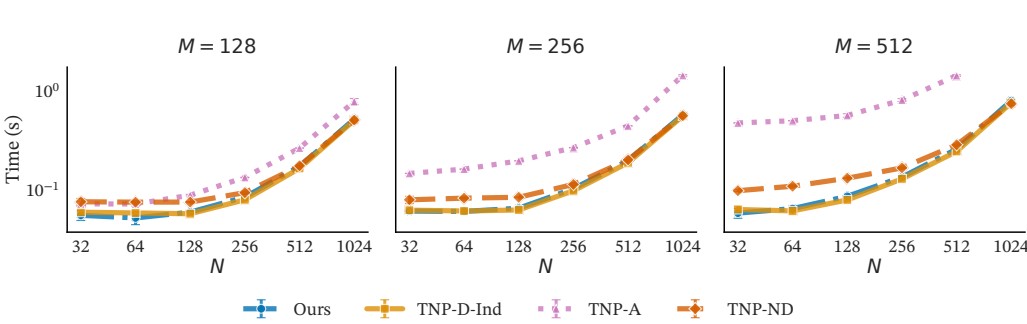

Figure A11: Training step time vs. number of target points $M$ using the standard PyTorch attention backend (FlashAttention disabled). Batch size $B = 128$.

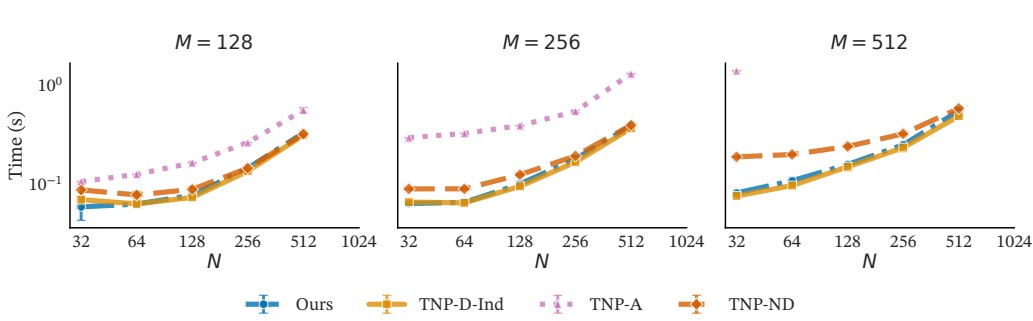

Figure A12: Training step time vs. number of target points $M$ using the standard PyTorch attention backend (FlashAttention disabled). Batch size $B = 256$.

## C.6 MEMORY USAGE

Figure A13 reports peak GPU memory consumption during autoregressive sampling as a function of context size $N$ across different batch sizes $B$. Our method maintains consistently low memory usage across all configurations, requiring 6–7× less VRAM than TNP-D-AR and TNP-A at large context sizes ($N = 1024$). This efficiency stems from our fixed-size buffer mechanism: while autoregressive baselines must cache representations that grow with context size and batch size, our method only caches buffer representations of size $K$, independent of the batch. TNP-D-Ind and

TNP-ND show lower memory usage but, as demonstrated in the main text, cannot capture complex predictive dependencies.

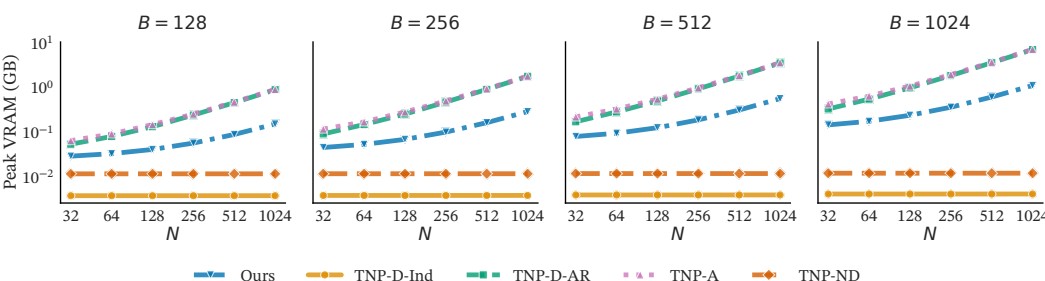

Figure A13: Peak GPU memory usage during sampling as a function of context points $N$ for different batch sizes $B$. Our method scales efficiently due to its fixed buffer size, using substantially less memory than expressive autoregressive baselines.

## D   EXPERIMENTAL DETAILS

### D.1   MODEL CONFIGURATION

In our paper, we use MLP to map context pairs, buffer pairs, or target points to tokens. Then a transformer is applied to the sequence of tokens. We used mixture-of-Gaussian (GMM) head as our main head distribution (more expressive than a single Gaussian head, as demonstrated in Appendix E). In general, we train all models (except the tabular model; see Appendix D.4 for details) with the following settings.

**Training configurations.**

- Optimizer: Adam with learning rate $1 \times 10^{-4}$ (unless stated otherwise), $\beta = (0.9, 0.999)$, no weight decay. For TNP w/ buffer, we use the same settings, but apply weight decay of $0.01$ for stability.
- Scheduler: Cosine schedule with warmup; warmup ratio $0.1$ for all experiments. For TNP w/ buffer, we use a warmup ratio of $0.05$.
- Training loop: 32 epochs.

**Embedder.**   We use a 3-layer MLP with 256 hidden layer dimension and 128 output dimension. There is a skip connection between the first linear layer and the MLP output.

**Transformer backbone.**   This has 6 layers of transformer encoder modules, each with a multi-head attention of 4 heads and dimension 128 followed by an MLP feedforward of 2 layers, dimension $128 \rightarrow 256 \rightarrow 128$. This is the transformer attending context, buffer, and target set (Appendix A and Appendix B).

**Prediction head.**   Note first that different distribution heads involve individual parameterization structures. Therefore, another layer of distribution-specific NNs is required to process the above transformer outputs. This NN module is considered part of the distribution head (the $\psi$ in Appendix A and Appendix B).

For our method, **TNP-D**, and **TNP-A**, the head consists of 2 layers of MLP with dimension $128 \rightarrow 256 \rightarrow 3 * D_y * N_{\text{components}}$, where $D_y$ is the output dimension of the problem and $N_{\text{components}}$ is the number of Gaussian components. The MLP output is then chunked into weights, means, and standard deviations (of the same shape) which parameterize the GMM, and the outputs are sampled in parallel for $D_y > 1$. We set $N_{\text{components}} = 20$ for all tasks except for EEG where $N_{\text{components}} = 8$.

For **TNP-ND**, we use the setting from Nguyen & Grover (2022), where the targets are mapped to a mean and a Cholesky matrix, which parameterize the multivariate Gaussian. The mean of each target is mapped by an MLP with dimension $128 \rightarrow 256 \rightarrow D_y$. The Cholesky matrix requires two steps: (i) the target tokens (conditioned on context via the above transformer backbone) are first decoded into $H \in \mathbb{R}^{M \times 20}$ by another 3-layer transformer (no positional encoding, 4 heads, each layer with dimension 128 and MLP $128 \rightarrow 256 \rightarrow 128$, no mask) and then an MLP projector ($128 \rightarrow 256 \rightarrow 20$); (ii) the Cholesky matrix is taken as $L = \text{lower}(HH^T)$.

**Trained model selection.** We track the loss value in each epoch as we train the models. The parameters with the best loss value on the validation set are selected for the evaluations on a separate test set.

## D.2 DATASETS

**Gaussian Process (GP) Functions.** As a first toy case, we test on GP functions (see Rasmussen & Williams 2006 for details of GPs). In this example, a batch contains 128 functions of one dimensional inputs ($D = 1$) and one dimensional observations ($D_y = 1$). The inputs are sampled from interval $[-2, 2]$ using the Sobol sequence. For each batch, we first sample a kernel class from squared-exponential (RBF), Matérn-$3/2$, Matérn-$5/2$ with probabilities 0.4, 0.3, and 0.3, respectively. Conditional on the chosen class, each function receives its own kernel hyperparameters: the variance $\sigma_f^2 \sim \text{Uniform}[0.5, 1.5]$ and the lengthscale $\ell \sim \text{Uniform}[0.1, 1]$, broadly covering diverse classes of functions of amplitude around 1. We then sample functions from $\mathcal{GP}(0, \boldsymbol{k})$, where $\boldsymbol{k}$ represents the sampled kernels, and add i.i.d. Gaussian observation noise with variance $10^{-5}$. The resulting values are randomly partitioned into context, buffer, and target sets. Note that within a batch the kernel class is fixed, whereas the hyperparameters are sampled independently for each function.

During training, we sample the context set size $N$ between 4 and 192 with a maximum buffer size of 16.

**Sawtooth Functions.** The second example is the non-Gaussian sawtooth functions (Bruinsma et al., 2023). In this example, a batch contains 128 functions of one dimensional inputs ($D = 1$) and one dimensional observations ($D_y = 1$). The inputs are sampled from interval $[-2, 2]$ using the Sobol sequence. An input $\mathbf{x}$ and output $y$ follows:

$$y(\mathbf{x}) = y_{\text{nonoise}}(\mathbf{x}) + \epsilon,$$
$$y_{\text{nonoise}}(\mathbf{x}) = (\omega(\langle u, \mathbf{x} \rangle - \phi)) \bmod 1,$$

where $u \in \mathbb{R}^D$ is a direction sampled uniformly from the unit sphere via $u = g/\|g\|_2$ with $g \sim \mathcal{N}(0, I_D)$; $\omega, \phi$, and $\epsilon$ denote the frequency, phase offset, and additive noise, respectively; and the parameters are drawn independently as $\omega \sim \text{Uniform}[3, 5]$, $\phi \sim \text{Uniform}[0, 1]$, and $\epsilon \sim \mathcal{N}(0, \sigma^2)$ with noise scale $\sigma \sim \text{Uniform}[0.05, 0.1]$.

During the training, we sample $N$ between 8 and 128 and the maximum number of buffer is 16.

**Electroencephalogram (EEG).** The dataset contains $11,520$ trials of 122 subjects from 7 correlated channels with 256 time points each. The output channels are individually standardized to zero mean and unit variance. We randomly select 10 for the test set, reserve 10 for cross-validation, and the remaining for the train set. This leaves 7802 trials for the training and 896 for testing.

During the training, the trials are replicated for 200 times and shuffled. Each batch contains 32 trials sampled from the shuffled set. We select between 4 and 192 of the 256 time points to be context points, 32 buffer points, with the remaining being target points. Each batch has a fixed size of context set.

We evaluate on both interpolation (random masking) and forecasting (temporal masking) tasks using the test subjects. The test set splits the 256 time points into context and target. For interpolation, we sample the specified number of context and target points from the full time sequence (Appendix E). For forecasting, we take the first $N$ points as context set and the consecutive $M$ points as target set. Forecasting with $N = 192$ context and $M = 64$ target sets involves the full sequence.

**Multisensory causal inference model dataset.** In the last example, we adopt one of the multisensory causal inference models described in Liu et al. (2025) to build a simulator, which we then use to generate training data (full setup and generation procedure, as well as a description of the experiment, are provided in Appendix D.3). The inputs $\mathbf{x}$ correspond to the experimentally manipulated variables of the study, namely $r_{\text{type}}, s_A, s_V$ and $V_{\text{level}}$, where $r_{\text{type}}$ denotes the task type (auditory vs visual localization), $s_A$ and $s_V$ are the true locations of auditory and visual cues presented to human participants, and $V_{\text{level}}$ the level of noise applied to visual cues. We first generate sets of input points for the simulator to obtain the outputs $y$, which represent the predicted responses.

For training, we construct two datasets from the simulator with different values of $\rho$, a variable of the model regulating the level of recalibration of the auditory perceptual range (with $\rho = 1$ representing no recalibration and $\rho = 4/3$ representing a full recalibration to the visual range, see Appendix D.3 for more details), and train two separate models for each setting. We sample $N$ between 0 and 400, fix $N_t = 256$, and set the buffer size to a maximum of 16. For the zero-context case, we introduce one "dummy point", to indicate the absence of context to the model. During evaluation, we use the publicly available dataset obtained from the experiment described in Liu et al. (2025). [8] For each of the 15 participants in the study, we extract two non-overlapping subsets of experimental data of 400 trials each. We do so by stratifying on the joint levels of $V_{\text{level}} \in \{0, 1, 2\}$ and $r_{\text{type}} \in \{0, 1\}$ (more details on these variables below), and extracting the two sets such that (i) within each split the six $(3 \times 2)$ strata are represented as evenly as possible, and (ii) the per-stratum counts are matched between splits. This yields 30 datasets overall (2 per participant).

For details of the real experiments and the complete data generation setup in the simulator, see Appendix D.3.

**Tabular foundation model.** We pretrain a task-agnostic tabular model on synthetic data and evaluate it on three UCI datasets. This larger model uses a dedicated training procedure; architectural and training details are in Appendix D.4, and the evaluation protocol is in Appendix D.5.

## D.3 Multisensory causal inference model and experiment details

To probe our method's suitability for Bayesian model comparison, we consider a computational neuroscience study investigating multisensory causal inference, described in Liu et al. (2025).

### D.3.1 Original neuroscience experiment

**Stimuli and procedure.** In this work, we take into account a subset of the experimental data obtained from 15 human participants who, at each experimental trial, were asked to perform one of two localization tasks, which the authors refer to as bisensory visual (BV) and bisensory auditory (BA) localization. In both cases they were presented with an auditory cue, located at an angle uniformly sampled among $\{-15°, -10°, -5°, 0°, 5°, 10°, 15°\}$ from the participant, and a visual one, either at the same location as the auditory one ($\approx 1/2$ of trials) or at an angle uniformly sampled between $-20°$ and $20°$. They were either asked to report the location of the visual (BV) or the auditory (BA) stimulus on a screen. Here we call those locations $s_V$ and $s_A$, respectively. The level of noise $V_{\text{level}}$ associated with the visual stimulus location was experimentally manipulated by modifying the size of the stimulus itself. In practice, this meant presenting a small ($V_{\text{level}} = 0$; $\approx 1/3$ of trials), medium ($V_{\text{level}} = 1$; $\approx 1/3$ of trials) or large ($V_{\text{level}} = 2$; $\approx 1/3$ of trials) visual stimulus.

Each participant completed a total of 1000 trials.

**Cognitive models.** Here we focus on two versions of the "vanilla" model described in the original paper (Liu et al., 2025). On each trial, the participant is assumed to believe the two stimuli could come from either a common ($C = 1$) or different ($C = 2$) source, assigning a fixed prior probability $p(C = 1) = p_{\text{same}}$ to the former case. Regardless of this, the participant has Gaussian priors over stimuli locations $p(s_A) = \mathcal{N}(s_A \mid 0, \sigma_S^2)$ and $p(s_V) = \mathcal{N}(s_V \mid 0, \sigma_S^2)$.

A key assumption of the model is that participants do not have direct access to the true location of the stimuli, but only to noisy auditory and visual percepts, a common feature in Bayesian models of perception (Knill & Pouget, 2004). These percepts are modeled as $x_A = \rho(s + \varepsilon_A)$ and $x_V = s + \varepsilon_V$

---

[8] https://github.com/LSZ2001/Audiovisual-causal-inference

respectively in case of a common source, and $x_A = \rho(s_A + \varepsilon_A)$ and $x_V = s_V + \varepsilon_V$ in case of separate sources. Here $s = s_A = s_V$ represents their common location when $C = 1$, while $\varepsilon_A \sim \mathcal{N}(0, \sigma_A^2)$ and $\varepsilon_V \sim \mathcal{N}(0, \sigma_V^2)$ represent the auditory and visual perceptual noise. While $\sigma_A$ is assumed to be fixed, $\sigma_V$ can assume three separate values ($\sigma_V^{(\text{low})}, \sigma_V^{(\text{med})}, \sigma_V^{(\text{high})}$) based on the (experimentally manipulated) size of the visual stimulus $V_{\text{level}}$. Finally, $\rho$ represents a "recalibration" factor to account for the fact that the range of auditory stimuli ($30°$) is different from that of visual ones ($40°$). In our experiment, this is the factor that differentiates the two models we set out to compare: in the first, we set $\rho = 1$; in the second, we set $\rho = 4/3$ (thus re-mapping auditory percepts to the same scale as visual ones).

Here we describe a BA trial, but the following is easily generalizable to BV ones. When asked about the location of the auditory stimulus, participants are assumed to consider both scenarios (common vs different sources) by evaluating

$$p(s \mid C = 1) = p(s \mid x_A, x_V, \sigma_A, \sigma_V, \sigma_S),$$
$$p(s_A \mid C = 2) = p(s_A \mid x_A, \sigma_A, \sigma_S),$$

as well as

$$p(C \mid x_A, x_V, \sigma_A, \sigma_V, \sigma_S, p_{\text{same}}).$$

The final estimate $\hat{s}_A$ of the location is then inferred by weighting the two hypotheses (common vs separate sources) by their posterior probability, so

$$\hat{s}_A = p(C = 1 \mid x_A, x_V, \sigma_A, \sigma_V, \sigma_S, p_{\text{same}}) \int_{-\infty}^{\infty} s \cdot p(s \mid C = 1) ds +$$
$$p(C = 2 \mid x_A, x_V, \sigma_A, \sigma_V, \sigma_S, p_{\text{same}}) \int_{-\infty}^{\infty} s_A \cdot p(s_A \mid C = 2) ds_A. \tag{7}$$

Finally, the response of the participant is modeled as $y \sim \mathcal{N}(\hat{s}_A, \sigma_M^2)$ with a probability of $1 - \lambda$, and $y \sim \text{Uniform}[-45, 45]$ with a probability of $\lambda$. Here $\lambda$ represents the "lapse rate", or the probability of a participant being distracted/disengaged and giving a random answer (which we fix at 0.02), while $\sigma_M$ represents motor noise.

Both models thus have 7 free parameters, which we re-parametrize as $\log \sigma_V^{(\text{low})}, \log \sigma_V^{(\text{med})}, \log \sigma_V^{(\text{high})}$, $\log \sigma_A, \log \sigma_S, \log \sigma_M$ and $\text{logit} p_{\text{same}}$ for the purposes of simulation and model-fitting.

### D.3.2 SIMULATION

For training all models, we produce $\sim 1.5$ millions synthetic datasets. In what follows we go through the simulation of a single trial. As trials are independent from one another, generating more of them simply involves repeating this process.

**Stimuli.** Following the setup used in Liu et al. (2025), we sample $s_A \sim$ Uniform$\{-15, -10, -5, 0, 5, 10, 15\}$ and $C \sim$ Uniform$\{1, 2\}$. Then we either sample $s_V \sim$ Uniform$[-20, 20]$ as a continuous variable (if $C = 2$) or we set $s_V = s_A$ (if $C = 1$). We then sample $V_{\text{level}} \sim$ Uniform$\{0, 1, 2\}$, representing the perceptual noise associated with $s_V$. This regulates whether $\sigma_V = \sigma_V^{(\text{low})}, \sigma_V = \sigma_V^{(\text{med})}$ or $\sigma_V = \sigma_V^{(\text{high})}$.

Finally, we sample $r_{\text{type}} \sim$ Uniform$\{0, 1\}$, representing the task (BV if $r_{\text{type}} = 0$, BA if $r_{\text{type}} = 1$).

**Parameters.** For each synthetic dataset, the prior generative distributions for the 7 free parameters are Gaussians truncated at two standard deviations above and below the mean. We use $\mathcal{N}_{\text{truncated}}(\mu, \sigma^2)$ to denote such distributions, with $\mu$ being the mean and $\sigma$ the standard deviation. Similarly to empirical Bayes approaches (Murphy, 2023), we use maximum-likelihood estimates of the individual participants' parameters from Liu et al. (2025) as a guide for setting these priors, so

as to generate realistic parameter ranges. The parameter distributions we use in this work are:

$$\log\sigma_V^{(\text{low})} \sim \mathcal{N}_{\text{truncated}}(0, 1.5^2);$$
$$\log\sigma_V^{(\text{med})} \sim \mathcal{N}_{\text{truncated}}(\log\sigma_V^{(\text{low})} + 1, 1^2);$$
$$\log\sigma_V^{(\text{high})} \sim \mathcal{N}_{\text{truncated}}(\log\sigma_V^{(\text{med})} + 0.75, 0.5^2);$$
$$\log\sigma_A \sim \mathcal{N}_{\text{truncated}}(1.75, 0.5^2);$$
$$\log\sigma_S \sim \mathcal{N}_{\text{truncated}}(2.5, 1^2);$$
$$\log\sigma_M \sim \mathcal{N}_{\text{truncated}}(0, 0.5^2);$$
$$\text{logit}\,p_{\text{same}} \sim \mathcal{N}_{\text{truncated}}(1.5, 1.5^2).$$

Note that $\log\sigma_V^{(\text{low})}$, $\log\sigma_V^{(\text{med})}$, and $\log\sigma_V^{(\text{high})}$ are not independent from each other, but carry the assumption that in most cases $\log\sigma_V^{(\text{low})} \lesssim \log\sigma_V^{(\text{med})} \lesssim \log\sigma_V^{(\text{high})}$, which reflects the intent of the experimental manipulation of $V_{\text{level}}$.

**Responses.**  Here we describe a scenario in which $r_{\text{type}} = 1$ (BA trial), but the process is the same for $r_{\text{type}} = 0$. In simulating the responses, we follow the hierarchical structure specified by the model. First we computed the sensory percepts $x_A = \rho(s_A + \varepsilon_A)$ and $x_V = s_V + \varepsilon_V$ by sampling $\varepsilon_A \sim \mathcal{N}(0, \sigma_A^2)$ and $\varepsilon_V \sim \mathcal{N}(0, \sigma_V^2)$. We then evaluate $\hat{s}_A$ (recall we are considering a BA trial) as in Eq. (7), and sample the final response as either $y \sim \mathcal{N}(\hat{s}_A, \sigma_M^2)$ or $y \sim \text{Uniform}[-45, 45]$, with a probability regulated by the lapse rate $\lambda$ (which we set to 0.02, see above).

### D.3.3 GROUND-TRUTH ACQUISITION

Here we describe how we obtained our log marginal likelihood (LML) estimates (in the form of lower bounds, see below), which we then use as ground-truth to compare our approach to baselines.

**Problem setting.**  Fitting the cognitive model to a dataset involves finding the posterior over model parameters given empirical data and model

$$p(\boldsymbol{\theta} \mid \mathbf{y}, \mathbf{X}, \rho) = \frac{p(\mathbf{y} \mid \boldsymbol{\theta}, \mathbf{X}, \rho)p(\boldsymbol{\theta})}{p(\mathbf{y} \mid \mathbf{X}, \rho)}, \tag{8}$$

where

$$\boldsymbol{\theta} = \{\log\sigma_V^{(\text{low})}, \log\sigma_V^{(\text{med})}, \log\sigma_V^{(\text{high})}, \log\sigma_A, \log\sigma_S, \log\sigma_M, \text{logit}\,p_{\text{same}}\},$$
$$\mathbf{X} = \{s_A^{(t)}, s_V^{(t)}, V_{\text{level}}^{(t)}, r_{\text{type}}^{(t)}\}_{t=1}^{400},$$

and

$$\mathbf{y} = \{y^{(t)}\}_{t=1}^{400}.$$

Here $t$ represents the trial number within the dataset (recall we are using data splits of 400 trials each, see Appendix D.2), and we set $p(\boldsymbol{\theta})$ to the truncated Gaussians we use for sampling the parameters in our simulation (see Appendix D.3.2), with probability density of values beyond the truncation boundaries set to a "floor value" of $\mathcal{N}(5 \mid 0, 1)$.

While the posterior over parameters is often instrumental in answering scientific questions, the crucial quantity we are interested in estimating is the model evidence (also called marginal likelihood) $p(\mathbf{y} \mid \mathbf{X}, \rho)$ (i.e., the denominator in Eq. (8)), as it represents a straightforward metric for model selection. In fact, assuming a flat prior over models $p(\rho = 1) = p(\rho = 4/3) = 0.5$, the model evidence as a function of $\rho$ represents the unnormalized posterior over models.

**Stacking Variational Bayesian Monte Carlo.**  To compute a reliable estimate of the marginal likelihood to use as our ground-truth, we use *Stacking Variational Bayesian Monte Carlo* (S-VBMC, Silvestrin et al., 2025). This is a principled approach to merge ("stack") approximate posteriors generated by a set of independent runs of its parent algorithm, Variational Bayesian Monte Carlo (VBMC, Acerbi, 2018; 2020). This is done in a simple post-processing step, which has been shown to greatly improve the approximate posterior quality in a variety of challenging settings. In addition to a posterior distribution, S-VBMC outputs an estimate of the evidence lower bound (ELBO),

which, as the name suggests, is a lower bound on the (log) model evidence (Blei et al., 2017), the quantity we are interested in for model comparison. As the approximation of the posterior approaches the true one, this quantity gets closer to the true model evidence, with equality when the approximation is perfect. As S-VBMC proved very effective in computational neuroscience problems (Silvestrin et al., 2025), including one very similar to the one considered here (Acerbi et al., 2018), we deem it a suitable method for estimating a lower bound on model evidence to use as a ground-truth.

While an in-depth description of S-VBMC and VBMC is beyond the scope of this work (an interested reader should refer to the original papers cited above), in the following paragraphs we briefly report details of our implementation of both.

*VBMC implementation details.* To obtain an approximate posterior, the Python implementation of VBMC (Huggins et al., 2023) requires absolute and plausible upper and lower bounds for each parameter. We use the sampling bounds defined in Appendix D.3.2 as absolute bounds, and replicate the process considering 1.5 standard deviations (as opposed to 2) from the mean to establish the plausible ones.

Another required input is a target density function (i.e., the unnormalized posterior), for which we use the numerator of Eq. (8), $p(\mathbf{y} \mid \boldsymbol{\theta}, \mathbf{X}, \rho)p(\boldsymbol{\theta})$. We do this both with $\rho = 1$ and $\rho = {}^4/_3$, representing the two models we set out to compare.

Finally, VBMC requires a starting point in the parameter space, which we uniformly sample between plausible bounds independently for each inference run.

*S-VBMC implementation details.* After obtaining 20 converging VBMC runs for each of our 30 datasets (2 for each of the 15 participants, see Appendix D.2) for both models, we stack the resulting posteriors with S-VBMC. We maintain the default settings, therefore the only inputs required are the VBMC runs themselves. With this, we obtain a total of 60 "stacked" ELBOs (two per each dataset, corresponding to our two competing models) to use as ground-truth.

## D.4 Tabular Model Details

This section describes the TabICL model and explains how the training dataset was generated. Notably, the base architecture used for this tabular data example is different from the one used in the other experiments, highlighting the broad applicability of our method.

### D.4.1 Architecture

**Set encoder.** We reuse the first two stages of TabICL (Qu et al., 2025) without modification: the distribution-aware column processor ($\text{TF}_{\text{col}}$, implemented with induced self-attention blocks) followed by the context-aware row-wise transformer ($\text{TF}_{\text{row}}$) with RoPE. Scalars are mapped by a $1 \rightarrow 128$ linear layer; each column is then processed across rows by an ISAB stack (Lee et al., 2019) with three blocks, four heads, 128 inducing points, feed-forward hidden dimension of 256. The row-wise encoder has three layers with four heads, feed-forward hidden dimension of 256, and RoPE base 100,000. We prepend two `[CLS]` tokens per row and concatenate their outputs, yielding a 256-dimensional row embedding ($2 \times 128$). We use at most ten features per table.

**Tokenization and additive target encoding.** The set encoder produces one row token per sample for context, buffer, and target rows (dimension 128; only selects the subset of the vector corresponding to the `[CLS]` token dimensions). Context and buffer tokens receive the target value *additively* via a small target encoder (linear $1 \rightarrow 128$. Buffer tokens also receive a learned positional embedding indicating their autoregressive index (up to 32 positions). This keeps labels additive, lets us compute the set encoder once, and makes the buffer explicit at the token level.

**Dataset-wise ICL with a buffered mask.** On top of these tokens we run a dataset-wise transformer with twelve layers and four heads, model width 128, and feed-forward size 256. The attention mask is the only architectural change relative to TabICL: context attends bidirectionally and is read-only at inference; the buffer uses strictly causal self-attention; target queries attend to the cached context and to the causal prefix of the buffer; there are no edges into context from buffer or targets. The maximum buffer size is 32 tokens and we query 512 targets per task.

**Prediction head.** Predictions use a GMM head with 20 components and a minimum standard deviation of $10^{-3}$.

**Caching.** The column and row set encoder is computed once for all rows. During autoregressive decoding we cache keys/values for the context once and update only the buffer cache, so the same context cache is reused across parallel generations.

### D.4.2 DATA GENERATION AND PREPROCESSING

**SCM prior and task family.** We generate datasets with the MLP-based *structured causal model* (SCM) prior in the style of Hollmann et al. (2023), following the dataset-wise, set-encoded regime of TabICL (Qu et al., 2025). Concretely, we first sample a DAG with layered (MLP-style) connectivity and then define each variable $c$ as $c = f(\mathrm{Pa}(c)) + \varepsilon$, where $\mathrm{Pa}(c)$ are its parents, $f$ is a small MLP with nonlinearity, and $\varepsilon$ is independent noise. Unless stated otherwise, we sample the feature dimension $d \in [1, 10]$, and per-task context sizes $N \in [8, 1024]$; targets are continuous responses with dataset-specific noise levels. The cause sampler follows the TabPFN prior (including mixed marginals); the SCM therefore yields columns that may be non-Gaussian or discrete at source, which we handle with the TabICL preprocessing described below.

**Sampling of task partitions.** For each generated dataset we draw a random partition $(\mathcal{C}, \mathcal{B}, \mathcal{T})$ with $N \sim \mathrm{Uniform}\{8, \ldots, 1024\}$, buffer capacity fixed at $K = 32$, and target count $M = 512$. Per batch, we fix $(d, N, K, M)$ across tasks to avoid padding and stack samples directly.

**Preprocessing.** We adopt the TabICL *PreprocessingPipeline* and fit it on context features only. The fitted transform is then applied to context, buffer, and target features. Regression targets are standardized using context statistics, i.e., $\tilde{y} = (y - \mu_{y,\mathcal{C}})/\sigma_{y,\mathcal{C}}$, and the same $(\mu, \sigma)$ are used for buffer and targets. No missing values are synthesized by the SCM generator.

*Summary of preprocessing pipeline.* We use a three-stage, per-column pipeline following Qu et al. (2025): (i) standard scaling; (ii) normalization (`power`, i.e., Yeo–Johnson); and (iii) outlier handling via a $z$-score threshold $\tau = 4.0$. At transform time, values outside the fitted range are clipped to the training (context) min/max before normalization, mirroring TabICL's behavior.

### D.4.3 TRAINING PROCEDURE

We train with AdamW (learning rate $1 \times 10^{-4}$, $\beta$=(0.9, 0.95), weight decay 0.0), batch size 64 datasets per step, gradient clipping at 0.5, and dropout 0.0 throughout the backbone. Mixed-precision training uses AMP with `bfloat16`. All runs use `float32` tensors at the data interface. A cosine schedule with warmup is used (`cosine_with_warmup`); `warmup_steps` = 2000 takes precedence over the nominal `warmup_ratio` = 0.20; `num_cycles` = 1. Automatic mixed precision is enabled with `amp_dtype=bfloat16`. Each training step draws a batch of 64 independent tasks (datasets) with feature dimension $d$ sampled from $\{1, \ldots, 10\}$ and context size $N$ from $\{8, \ldots, 1024\}$; buffer size and target count are fixed at $K$=32 and $M$=512. Training is capped at `max_steps` = 160,000, i.e., one epoch effective duration. This corresponds to approximately $64 \times 160,000 = 10.24$ million synthetic tasks seen during pretraining. The global data seed is 123. We trained the model on a single NVIDIA A100 80 GB GPU for approximately 3 days.

## D.5 EVALUATION DETAILS

In this paper log-likelihood values are always averaged (LL divided by the number of target points $M$).

**GP & Sawtooth functions.** We evaluate likelihood values over 1024 functions, each repeated 4 times with models trained on different seeds and context sizes $N = 8, 16, 32, 64, 128$ (statistics of $1024 \times 4 \times 5$ evaluations). Each likelihood evaluation is an average of 128 permutations (log averaged likelihood). In other words, we have $1024 \times 4 \times 5$ averaged likelihoods, and each averaged value merges 128 orders of the target set.

Table A1: **Head comparison on synthetic function.** We compare average log-likelihood (↑) results on our main GMM head and on standard Gaussian distribution head.

| | TNP-D | | TNP w/ buffer | | |
| | AR | Ind | $K$=16 | $K$=4 | $K$=1 |
|---|---|---|---|---|---|
| GP ($M = 16$) | 2.57 (0.020) | 2.22 (0.022) | 2.51 (0.019) | 2.55 (0.019) | 2.56 (0.019) |
| GP ($M = 128$) | 3.29 (0.013) | 2.15 (0.022) | 3.27 (0.013) | 3.28 (0.013) | 3.29 (0.013) |
| Sawtooth ($M = 16$) | 1.05 (0.004) | 0.94 (0.005) | 1.00 (0.005) | 1.08 (0.004) | 1.09 (0.004) |
| Sawtooth ($M = 128$) | 1.15 (0.003) | 1.16 (0.003) | 1.15 (0.003) | 1.16 (0.003) | 1.16 (0.003) |
| | TNP-D-Gaussian | | TNP Gaussian w/ buffer | | |
| | AR | Ind | $K$=16 | $K$=4 | $K$=1 |
| GP ($M = 16$) | 2.50 (0.019) | 2.13 (0.023) | 2.48 (0.019) | 2.53 (0.019) | 2.53 (0.019) |
| GP ($M = 128$) | 3.23 (0.013) | 2.06 (0.023) | 3.25 (0.013) | 3.27 (0.013) | 3.27 (0.013) |
| Sawtooth ($M = 16$) | 0.96 (0.004) | 0.82 (0.006) | 0.85 (0.006) | 0.98 (0.004) | 0.99 (0.004) |
| Sawtooth ($M = 128$) | 1.10 (0.003) | 0.82 (0.005) | 1.10 (0003) | 1.11 (0.003) | 1.11 (0.003) |

**EEG data.** We train each model once with a fixed seed; the evaluations are over 896 trials from 20 subjects held out during training, each repeated with $N = 8, 16, 32, 64, 128, 192$. For the EEG forecasting, the target set consists of time points immediately after context points, and, in the main results (Table 1), the target set permutations are applied, as done in Bruinsma et al. (2023). We additionally demonstrate in appendix Table A3 that forecasting with permuted target set outperforms fixed sorted target. The number of permutations we apply is 128.

**Multisensory causal inference model.** We train one model for each setting of $\rho$ ($\rho = 1$ and $\rho = 4/3$). In the model selection scenario, the full 400-point dataset from each of the 30 batches is used as the target, and we evaluate the LML across all cases. This procedure is repeated 5 times, with 128 different sequence permutations per run. In the data prediction scenario, we first select the winning model from the model selection stage, and then compute log-likelihoods on the same 30 batches, each repeated with $N = 8, 16, 32, 64, 128, 256$. The results of both experiments are summarized in Table 2. Here we also use 128 permutation for all batches.

**Tabular foundation model.** We pretrain a task-agnostic tabular model on synthetic data (Appendix D.4) and evaluate it on three UCI datasets: Individual Household Electric Power Consumption[9], Gas Turbine CO and NOx Emission DataSet[10], Bike Sharing[11], Jena climate dataset[12], Power Consumption of Tetouan City[13], and California Housing Price[14].

For each dataset, we evaluate likelihood values over 16 randomly sampled subsets. The context and target sets are set to $N = 128, M = 32$. Each likelihood evaluation is an average of 128 permutations.

Table A2: **Average Log-likelihood (↑) results on synthetic functions and EEG example.** Supplementary results of Table 1 on larger target set and various deployed $K$. When $M > K$, we evaluate every $K$ targets once and perform AR for $M/K$ steps.

| | TNP-D | | TNP-ND | TNP-A |
|---|---|---|---|---|
| | AR | Ind | | |
| GP ($M = 16$) | 2.57 (0.020) | 2.22 (0.022) | 0.80 (0.082) | 2.24 (0.018) |
| GP ($M = 128$) | 3.29 (0.013) | 2.15 (0.022) | 2.27 (0.023) | 3.10 (0.012) |
| Sawtooth ($M = 16$) | 1.05 (0.004) | 0.94 (0.005) | -0.43 (0.008) | 0.98 (0.004) |
| Sawtooth ($M = 128$) | 1.14 (0.003) | 0.94 (0.005) | 0.39 (0.005) | 1.12 (0.003) |
| EEG-Int ($M = 16$) | 0.51 (0.013) | 0.36 (0.014) | 0.46 (0.011) | 0.58 (0.014) |
| EEG-Int ($M = 64$) | 0.88 (0.011) | 0.35 (0.014) | 0.50 (0.010) | 0.95 (0.012) |
| EEG-For ($M = 16$) | 1.07 (0.004) | -0.74 (0.008) | -0.04 (0.005) | 1.23 (0.003) |
| EEG-For ($M = 64$) | 1.12 (0.003) | -1.08 (0.007) | -0.23 (0.004) | 1.20 (0.003) |
| | TNP w/ buffer | | | |
| | $K$=16 | $K$=4 | $K$=1 | |
| GP ($M = 16$) | 2.51 (0.019) | 2.55 (0.019) | 2.56 (0.019) | |
| GP ($M = 128$) | 3.27 (0.013) | 3.28 (0.013) | 3.29 (0.013) | |
| Sawtooth ($M = 16$) | 1.00 (0.005) | 1.08 (0.004) | 1.09 (0.004) | |
| Sawtooth ($M = 128$) | 1.15 (0.003) | 1.16 (0.003) | 1.16 (0.003) | |
| EEG-Int ($M = 16$) | 0.52 (0.013) | 0.54 (0.014) | 0.54 (0.014) | |
| EEG-Int ($M = 64$) | 0.90 (0.011) | 0.91 (0.011) | 0.91 (0.011) | |
| EEG-For ($M = 16$) | 0.85 (0.004) | 1.17 (0.003) | 1.21 (0.003) | |
| EEG-For ($M = 64$) | 1.12 (0.003) | 1.18 (0.003) | 1.19 (0.003) | |

# E   ADDITIONAL LOG-PREDICTIVE DENSIITY RESULTS ON SYNTHETIC AND EEG TASKS

## E.1   PREDICTIVE POWER OF DIFFERENT HEADS

In this paper, we use GMM as our prediction head. We compare the predictive performance of GMM to standard Gaussian distribution head. In Table A1, GMM is able to achieve better predictive performance, particularly on the non-Gaussian Sawtooth functions.

## E.2   RESULTS OF LARGER $M$

As a supplementary results of Table 1, we evaluate log-likelihood values on a larger target set. For TNP w/ buffer, we evaluate $K$ points per Algorithm 2 and proceed to the next target subsets by conditioning on the context and evaluated points. This requires $M/K$ steps of evaluations. The results are reported in Table A2. As we decrease the number of buffer targets $K$ [15], the performance of our TNP w/ buffer becomes stronger, while more iterations (and thus computational time) are required.

---

[9] https://archive.ics.uci.edu/dataset/235/individual+household+electric+power+consumption

[10] https://archive.ics.uci.edu/dataset/551/gas+turbine+co+and+nox+emission+data+set

[11] https://archive.ics.uci.edu/dataset/275/bike+sharing+dataset

[12] https://www.kaggle.com/datasets/mnassrib/jena-climate

[13] https://archive.ics.uci.edu/dataset/849/power+consumption+of+tetouan+city

[14] https://www.kaggle.com/datasets/camnugent/california-housing-prices

[15] Note that when $K = 1$, our method is equivalent to standard TNP-D AR, as the actual number of points in the buffer is zero.

Table A3: **EEG forecasting w/ and w/o target set permutation.** The target set of EEG forecasting are the points immediately after the context set. Our main paper applies permutation to the target set while this table compares against forecasting of fixed temporal order (sorted).

| | TNP-D | | TNP-ND | TNP-A |
|---|---|---|---|---|
| | AR | Ind | | |
| EEG-For ($M = 16$) | 1.07 (0.004) | -0.74 (0.008) | -0.04 (0.005) | 1.23 (0.003) |
| EEG-For ($M = 16$, sorted) | 0.85 (0.005) | -0.74 (0.008) | -0.004 (0.005) | 1.14 (0.004) |
| EEG-For ($M = 64$) | 1.12 (0.003) | -1.08 (0.007) | -0.23 (0.004) | 1.20 (0.003) |
| EEG-For ($M = 64$, sorted) | 0.89 (0.005) | -1.08 (0.007) | -0.23 (0.004) | 1.16 (0.003) |
| | TNP w/ buffer | | | |
| | $K$=16 | $K$=4 | $K$=1 | |
| EEG-For ($M = 16$) | 0.85 (0.004) | 1.17 (0.003) | 1.21 (0.003) | |
| EEG-For ($M = 16$, sorted) | 0.76 (0.006) | 0.87 (0.005) | 1.09 (0.004) | |
| EEG-For ($M = 64$) | 1.12 (0.003) | 1.18 (0.003) | 1.19 (0.003) | |
| EEG-For ($M = 64$, sorted) | 0.78 (0.005) | 0.89 (0.004) | 1.11 (0.004) | |

Table A4: **Multisensory causal inference model selection extra results.** Supplement for Table 2 on model comparison case with extra evaluation on $K = 4$ and $R^2$ metrics for LML and $\Delta$LML.

| | TNP-D | | TNP-ND | TNP-A |
|---|---|---|---|---|
| | AR | Ind | | |
| LML RMSE ($\downarrow$) | 3.10 (0.005) | 86.96 (0.000) | 208.51 (0.041) | 4.75 (0.012) |
| $\Delta$LML RMSE ($\downarrow$) | 2.44 (0.008) | 36.18 (0.000) | 25.60 (0.023) | 3.29 (0.019) |
| LML $R^2$ ($\uparrow$) | 1.00 (0.000) | -0.43 (0.000) | -7.22 (0.003) | 1.00 (0.000) |
| $\Delta$LML $R^2$ ($\uparrow$) | 0.93 (0.001) | -14.47 (0.000) | -6.74 (0.014) | 0.87 (0.002) |
| | TNP w/ buffer | | | |
| | $K$=16 | $K$=4 | $K$=1 | |
| LML RMSE ($\downarrow$) | 3.56 (0.004) | 3.48 (0.002) | 3.47 (0.004) | |
| $\Delta$LML RMSE ($\downarrow$) | 2.60 (0.010) | 2.59 (0.009) | 2.59 (0.011) | |
| LML $R^2$ ($\uparrow$) | 1.00 (0.000) | 1.00 (0.000) | 1.00 (0.000) | |
| $\Delta$LML $R^2$ ($\uparrow$) | 0.92 (0.001) | 0.92 (0.001) | 0.92 (0.001) | |

### E.3 EEG FORECASTING W/ AND W/O TARGET PERMUTATION

In our main paper, the EEG forecasting task is evaluated with the permuted target set, following the procedure of Bruinsma et al. (2023). We repeat the experiment by forecasting the target of a fixed temporal order. In Table A3, we show that averaging over random target order as done in the paper, provides better overall performance.

## F ADDITIONAL MULTISENSORY CAUSAL INFERENCE MODEL RESULTS

As supplementary results to Table 2, we include additional metrics and evaluation settings. Specifically, for the model comparison task, we report the coefficient of determination ($R^2$) for both the LML and $\Delta$LML with respect to the ground-truth (see Table A4). For the data prediction task, we present results with a larger target size of $M = 128$ (see Table A5). In addition, for completeness, we evaluate both the model comparison and data prediction tasks with $K = 4$. With varying $K$, we observe little to almost no performance degradation compared to TNP-D AR, especially for the data prediction case.

## G ADDITIONAL TABULAR FOUNDATION MODEL RESULTS

We report results for an intermediate context size ($N = 256$) in Table A6. Consistent with our main findings, AR w/ buffer matches standard AR within standard errors across all tasks, while both autoregressive methods outperform independent predictions on forecasting tasks.

Table A5: **Multisensory causal inference model data prediction task normalized log-likelihood (↑) results.** Supplementary results of Table 2, with extra evaluation on $K = 4$ and on larger target set $M = 128$.

| | TNP-D | | TNP-ND | TNP-A |
|---|---|---|---|---|
| | AR | Ind | | |
| Pred LL ($M = 16$) | -2.76 (0.021) | -2.77 (0.025) | -3.12 (0.019) | -2.76 (0.024) |
| Pred LL ($M = 128$) | -2.71 (0.015) | -2.74 (0.016) | -3.17 (0.012) | -2.71 (0.015) |

| | TNP w/ buffer | | |
|---|---|---|---|
| | $K$=16 | $K$=4 | $K$=1 |
| Pred LL ($M = 16$) | -2.76 (0.024) | -2.76 (0.024) | -2.76 (0.024) |
| Pred LL ($M = 128$) | -2.71 (0.015) | -2.71 (0.015) | -2.71 (0.015) |

Table A6: **Average log-predictive density (↑) results on UCI datasets with TabICL ($N = 256$).** Results are reported as mean and standard error over 16 randomly sampled mini-datasets ($M = 32$) for interpolation (Int) and forecasting (For) tasks.

| | INTERMEDIATE CONTEXT REGIME ($N = 256$) | | | | | | | | | | |
|---|---|---|---|---|---|---|---|---|---|---|---|
| | Electric Cons. | | Gas Turbine | | Bike Sharing | | Tetouan | | Jena | | Cali. |
| | Int | For | Int | For | Int | For | Int | For | Int | For | Int |
| Independent | 1.65 (0.15) | 1.21 (0.30) | -0.44 (0.16) | -1.06 (0.32) | 2.38 (0.05) | 1.98 (0.11) | 0.56 (0.08) | -1.45 (0.44) | 2.03 (0.06) | 0.59 (0.18) | -0.55 (0.12) |
| Standard AR | 1.67 (0.14) | 1.57 (0.22) | -0.44 (0.16) | -0.73 (0.23) | 2.39 (0.05) | 2.24 (0.09) | 0.57 (0.08) | 0.39 (0.18) | 2.03 (0.06) | 1.45 (0.15) | -0.54 (0.12) |
| AR w/ buffer | 1.67 (0.14) | 1.56 (0.21) | -0.44 (0.16) | -0.76 (0.23) | 2.38 (0.05) | 2.23 (0.08) | 0.57 (0.08) | 0.28 (0.20) | 2.03 (0.06) | 1.30 (0.17) | -0.54 (0.12) |

# H ABLATIONS AND EXTRA EXPERIMENTS

## H.1 COMPARISON TO NON-PERMUTATION-INVARIANT TRANSFORMERS

To isolate the effect of permutation invariance in the context set, we replace our model with a plain autoregressive decoder Transformer that treats the context as a fixed input sequence. This sequential baseline performs substantially worse than our method in the GP task and across context sizes (see Fig. A14), indicating that explicitly maintaining permutation invariance over the context set – or at least part of it – is critical for performance.

## H.2 POSITIONAL EMBEDDINGS ABLATION

We also trained our method without positional embeddings in the buffer and performed evaluations with $M = K = 16$, as shown in Table A7, and observed no statistically significant difference in predictive performance. This aligns with findings in causal-transformer work showing that models can infer positional structure without explicit encodings (Haviv et al., 2022; Irie, 2024; Zuo et al., 2025). While not strictly required, positional embeddings may still support future extensions, such as scaling to larger buffer sizes via ALiBi (Press et al., 2022) or RoPE (Su et al., 2024).

Table A7: Average joint predictive log-density (↑) for positional embedding ablation on the GP task; reported as mean (SEM).

| TNP-D-AR | TNP-D-Ind | TNP-ND | TNP-A | Ours w/ pos. emb | Ours w/o pos. emb |
|---|---|---|---|---|---|
| 2.57 (0.02) | 2.22 (0.02) | 0.80 (0.08) | 2.24 (0.02) | 2.51 (0.02) | 2.51 (0.02) |

## H.3 NUMBER OF SAMPLES ORDER AVERAGING ABLATION

We study the effect of the number of sequence samples (permutations) used for order averaging. We report results on the multisensory causal inference task, where our method (with buffer size $K = 16$) is used to compute the LML of a dataset by averaging over multiple permutations. As shown in Fig. A15, increasing the number of permutations reduces the RMSE of the estimated joint

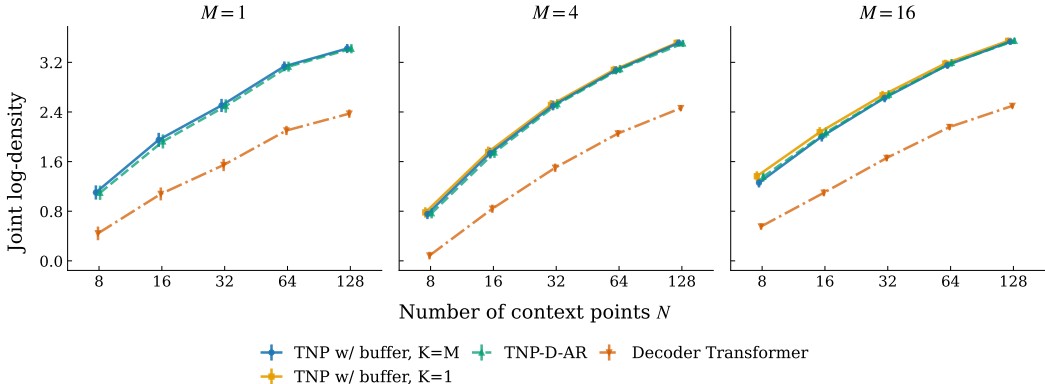

Figure A14: Average joint predictive log-density ($\uparrow$) on the GP regression task comparing the Decoder Transformer models with ours and gold standard TNP-D-AR on varying number of context points $N$ and number of targets $M = 1, 4, 16$.

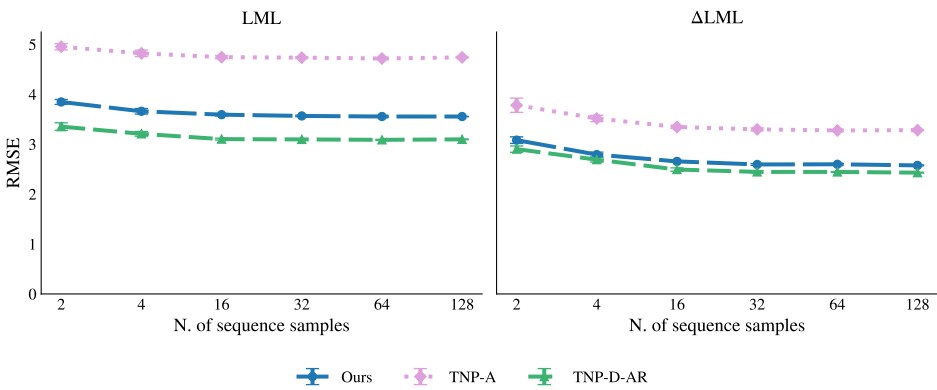

Figure A15: Average RMSE ($\downarrow$) on the LML (left) and $\Delta$LML (right) estimation in the multisensory causal inference task for different numbers of sample permutations.

density relative to the true joint at a rate comparable to existing autoregressive baselines, while our method remains significantly faster. This indicates that for our proposed method, order averaging does not introduce additional performance degradation relative to the gold standard baselines for a given number of permutations.

## H.4 EXTENSION TO LATENT BOTTLENECKED ATTENTIVE NEURAL PROCESSES MODEL

To assess the generality of the proposed autoregressive buffer, we integrate it into a perceiver-style Latent Bottlenecked Attentive Neural Processes (LBANP) architecture (Jaegle et al., 2021; Feng et al., 2023). The context set is first encoded into a fixed-size latent array, and the autoregressive buffer operates over targets on top of this latent array bottleneck. We evaluate this BNP with buffer model on the GP regression task with 4 and 16 targets. As shown in Fig. A16, the LBANP equipped with our autoregressive buffer matches or slightly outperforms a standard autoregressive deployment of the Perceiver architecture (when $K = 1$). This result is likely due to the fact that the buffer allows the model to *explicitly* represent the recent history of targets, bypassing the compressed representation of the context for immediate short-term dependencies, thus slightly enhancing predictive performance. These results demonstrate that our method extends naturally to bottlenecked / perceiver-style architectures, supporting its generality beyond full-attention models.

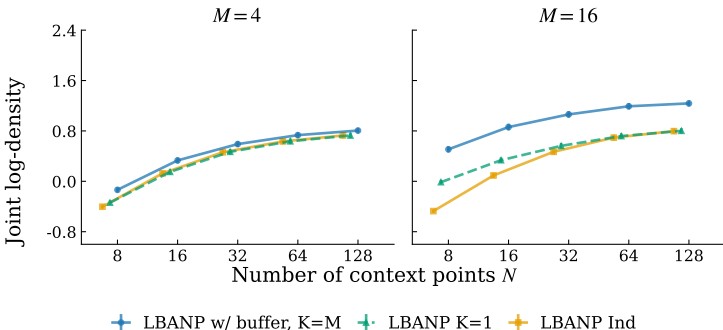

Figure A16: Autoregressive buffer extension on Latent Bottlenecked Attentive Neural Processes (LBANP) model. Average joint predictive log-density (↑) on GP with varying number of context $N$ and number of targets $M = 4, 16$.

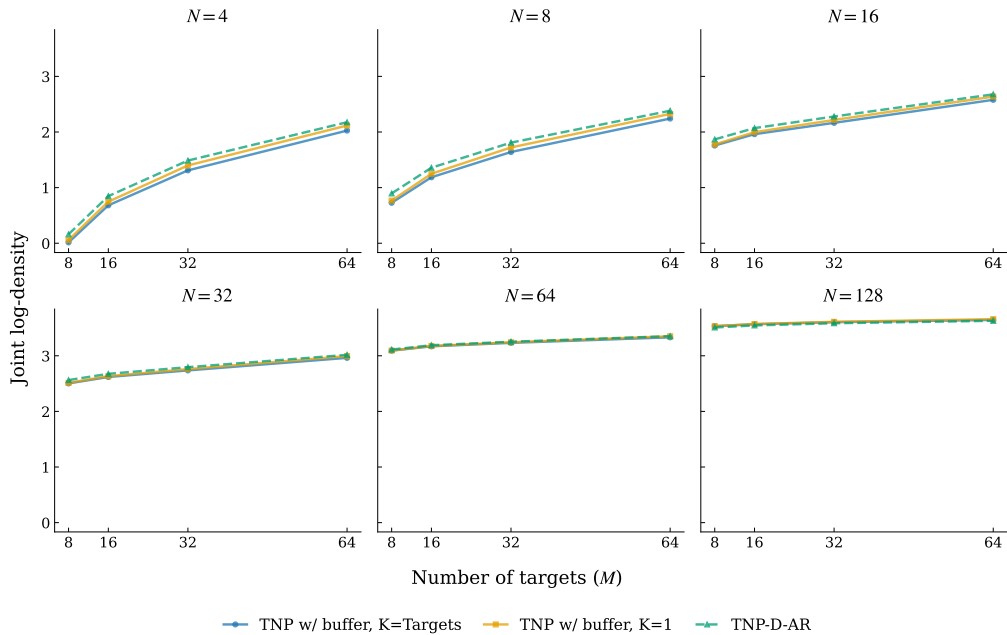

Figure A17: Average joint predictive log-density (↑) on the GP task across number of targets $M$ up to 64, with buffer size $K$ set equal to $M$, for varying numbers of context points $N$ from 4 to 128.

### H.5 BUFFER SIZE ABLATION

We evaluate the effect of the buffer size $K$ on the GP regression task, training a model with a maximum buffer size $K = 64$. As shown in Fig. A17, the performance of our method remains stable across this range and does not degrade relative to the autoregressive baseline, indicating that increasing $K$ up to 64 does not harm predictive quality.

## I USE OF LARGE LANGUAGE MODELS

**Idea generation and exploration.** We used Large Language Models (LLMs) in the early stages of this work to support idea generation, brainstorming, and the exploration of possible methodological

directions. LLMs were also employed for tasks such as identifying related work through web search and summarization, which helped us gain an initial overview of relevant literature.

**Coding assistant.** LLMs provided assistance with coding, primarily by generating boilerplate components of the codebase, visualization scripts, and test codes. They were also used for drafting parts of the implementation in PyTorch. All code produced or suggested by LLMs was carefully reviewed, verified, and modified where necessary to ensure correctness and reliability.

**Writing assistant.** Finally, LLMs were used in preparing the manuscript, particularly for refining clarity, conciseness, and grammatical correctness. They supported rephrasing and restructuring of text, helping us to communicate ideas more effectively while maintaining the accuracy and integrity of the content.

