# OpenReview forum: "Efficient Autoregressive Inference for Transformer Probabilistic Models"
_ICLR.cc/2026/Conference — ICLR 2026 Poster_

### Official Review · Reviewer_agRA · 2025-10-31

**Soundness:** 3
**Presentation:** 3
**Contribution:** 2
**Rating:** 6
**Confidence:** 3

**Summary:**

The paper introduces a causal autoregressive buffer for transformer-based amortized probabilistic models (NPs, PFNs, tabular FMs). The key idea is to encode the context once and keep it in a static cache; generated targets enter a causal buffer that attends to the context cache and prior buffer tokens, enabling joint dependencies without re-encoding the augmented set each step. Complexity drops from (O(K(N!+!K)^2)) to (O(N^2 + KN + K^2)), supporting batched AR sampling and single-pass joint likelihood evaluation. Experiments across synthetic, EEG, cognitive, and tabular tasks match standard AR accuracy while achieving speedups.

**Strengths:**

This is a practical architectural improvement that preserves set-conditioning benefits while unlocking efficient joint predictions typical of AR models. The idea of a role-aware attention mask (immutable context, causal buffer, no writes back to context) feels simple yet powerful, and widely applicable to TNPs/PFNs/tabular FMs. The potential for large efficiency gains in meta-learning scenarios with repeated sampling is significant.

- The factorization and masking constraints (R1–R4) are clearly stated; buffering semantics are sound. The complexity analysis matches the masking structure.
- Training uses a buffer-size curriculum with structured masks that let targets attend to context or a variable prefix of the buffer, aligning training and inference. The link to minimizing KL to posterior predictive under varying conditioning sets is consistent with PFN theory.
- The claims of accuracy comparison with standard AR and speedups are supported on small/medium contexts and (K< 32) (as per reported settings). The authors also acknowledge a degradation risk when target counts exceed the trained buffer size.

**Weaknesses:**

In the experiments, they are rather limited to moderate sizes of K and small/medium-sized contexts. I have a feeling that this may be limiting the characterization of the approach. Also, residual order dependence for likelihood evaluation requires averaging; I think an analysis on the order sensitivity is missing.

**Questions:**

- Is it possible to have tests on larger (K) regimes and very large contexts to map out failure boundaries, plus ablation on positional embeddings inside the buffer and ordering effects when approximating permutation invariance via order-averaging?
 - Is it possible to quantify order-averaging required to stabilize joint likelihoods with a metric measuring their cost/benefit?
- How do the authors decide on the size of (N)s? Is it there a experiment for emprically showing for large values of N, proposed method has compounding advantages?

---

> ### Author Response · Authors · 2025-11-24
> **Response to Reviewer agRA (Part 1 of 3)**
>
> We thank the reviewer for their comments that our proposed method is powerful, widely applicable, and has the potential to provide significant efficiency improvements for scenarios with repeated sampling requirements.
>
> > **Weaknesses:** In the experiments, they are rather limited to moderate sizes of K and small/medium-sized contexts. I have a feeling that this may be limiting the characterization of the approach. Also, residual order dependence for likelihood evaluation requires averaging; I think an analysis on the order sensitivity is missing.
>
> Thank you for raising these points. We respond to these in detail below, as they relate directly to your questions.
>
> > **Questions:**
>
> > **Q1a:** Is it possible to have tests on larger $(K)$ regimes and very large contexts to map out failure boundaries...
>
> Good point! To explore the larger $K$ regime, we provide below new results for the Gaussian process (GP) regression task on larger $K$, up to size 64, for $N = 16$ and $N = 128$. To push the large context regime, we show new results for the Tabular regression task, for $N = 256$ and up to $N = 1024$.
>
> In summary:
> * **Large $K$:** We find that the relative performance of our method does not degrade compared to the gold-standard AR method as the buffer size increases.
> * **Large $N$:** When stratifying by context size, we see that our method actually performs relatively *better* at higher context sizes than smaller ones. This points to the weakest setting for our method likely being very small context settings with a high number of target points.
> * **Failure Boundaries:** Overall, our method does not show degradation of performance up to buffer size $K = 64$, and its performance relative to other methods even *increases* at higher context sizes $N$. The other potential boundary to explore is $M > K$. Notably, as shown in Appendix E.2, if the number of targets $M$ exceeds the trained buffer size $K$, we can simply re-encode the buffer into the context and empty the buffer. While this requires an $O(N^2)$ operation and loses part of the advantages of our method, we only do it once every $K$ steps (unlike the baseline, which does it every step). Thus, the speedup and accuracy are preserved even in this regime.
>
> ---
>
> The results below, together with other intermediate results for other values of $N$, will be included in the revised paper (Table R1 as a figure; Table R2 and R3 as tables).
>
> Table R1: Average joint predictive log-density ($\uparrow$) on GP task across target counts up to 64 with number of context ($N$) of 16 and 128, reported as mean (SEM). These new results were obtained using models trained with a maximum buffer size $K$ of 64.
> | Model Configuration | 8 Targets | 16 Targets | 32 Targets | 64 Targets |
> | :--- | ---: | ---: | ---: | ---: |
> | **$N=16$, TNP w/ buffer, $K=1$** | 1.78 (0.04) | 2.00 (0.04) | 2.22 (0.04) | 2.64 (0.03) |
> | **$N=16$, TNP w/ buffer, $K=\text{Targets}$** | 1.76 (0.04) | 1.96 (0.04) | 2.16 (0.04) | 2.58 (0.03) |
> | **$N=16$, TNP-D-AR** | 1.87 (0.04) | 2.07 (0.04) | 2.28 (0.04) | 2.68 (0.04) |
> | **$N=128$, TNP w/ buffer, $K=1$** | 3.53 (0.03) | 3.57 (0.02) | 3.61 (0.02) | 3.66 (0.02) |
> | **$N=128$, TNP w/ buffer, $K=\text{Targets}$** | 3.53 (0.03) | 3.57 (0.02) | 3.60 (0.02) | 3.64 (0.02) |
> | **$N=128$, TNP-D-AR** | 3.51 (0.03) | 3.55 (0.03) | 3.58 (0.02) | 3.63 (0.02) |
>
>
> Table R2: Average joint predictive log-density ($\uparrow$) for various tabular foundation model tasks with number of context $N = 256$ reported as mean (SEM). These new $N = 256$ results, together with three additional tasks (Tetouan Power, Jena Climate, California Housing), were run during the rebuttal period. Int = Interpolation; For = Forecasting.
> | Task | Independent | Standard AR | AR w/ buffer |
> | :--- | :--- | :--- | :--- |
> | **Electric consumption** | | | |
> | Int | 1.65 (0.15) | 1.67 (0.14) | 1.67 (0.14) |
> | For | 1.21 (0.30) | 1.57 (0.22) | 1.56 (0.21) |
> | **Gas Turbine** | | | |
> | Int | -0.44 (0.16) | -0.44 (0.16) | -0.44 (0.16) |
> | For | -1.06 (0.32) | -0.73 (0.23) | -0.76 (0.23) |
> | **Bike Sharing** | | | |
> | Int | 2.38 (0.05) | 2.39 (0.05) | 2.38 (0.05) |
> | For | 1.98 (0.11) | 2.24 (0.09) | 2.23 (0.08) |
> | **Tetouan Power** | | | |
> | Int | 0.56 (0.08) | 0.57 (0.08) | 0.57 (0.08) |
> | For | -1.45 (0.44) | 0.39 (0.18) | 0.28 (0.20) |
> | **Jena Climate** | | | |
> | Int | 2.03 (0.06) | 2.03 (0.06) | 2.03 (0.06) |
> | For | 0.59 (0.18) | 1.45 (0.15) | 1.30 (0.17) |
> | **California House** | | | |
> | Int | -0.55 (0.12) | -0.54 (0.12) | -0.54 (0.12) |

---

> ### Author Response · Authors · 2025-11-24
> **Response to Reviewer agRA (Part 2 of 3)**
>
> > **Q1a (continued):** [Tables regarding Large Contexts $N=256$ and $N=1024$]
>
> Table R3: Average joint predictive log-density ($\uparrow$) for various tabular foundation model tasks  with number of context $N = 1024$ reported as mean (SEM). These new $N = 1024$ results, together with three additional tasks (Tetouan Power, Jena Climate, California Housing),  were run during the rebuttal period. Int = Interpolation; For = Forecasting.
> | Task | Independent | Standard AR | AR w/ buffer |
> | :--- | :--- | :--- | :--- |
> | **Electric consumption** | | | |
> | Int | 1.78 (0.06) | 1.78 (0.06) | 1.79 (0.06) |
> | For | 1.64 (0.18) | 1.70 (0.18) | 1.70 (0.18) |
> | **Gas Turbine** | | | |
> | Int | -0.01 (0.16) | -0.01 (0.16) | -0.01 (0.16) |
> | For | -0.60 (0.29) | -0.47 (0.27) | -0.48 (0.27) |
> | **Bike Sharing** | | | |
> | Int | 2.54 (0.05) | 2.54 (0.05) | 2.53 (0.05) |
> | For | 2.32 (0.07) | 2.40 (0.10) | 2.39 (0.06) |
> | **Tetouan Power** | | | |
> | Int | 0.36 (0.07) | 0.36 (0.07) | 0.36 (0.06) |
> | For | -1.12 (0.35) | -0.08 (0.22) | -0.12 (0.23) |
> | **Jena Climate** | | | |
> | Int | 2.01 (0.06) | 2.01 (0.06) | 2.01 (0.06) |
> | For | 1.56 (0.19) | 1.80 (0.13) | 1.64 (0.16) |
> | **California House** | | | |
> | Int | -0.44 (0.08) | -0.44 (0.08) | -0.44 (0.08) |
>
> ---
>
> > **Q1b:** ... plus ablation on positional embeddings inside the buffer...
>
> Thank you for this suggestion. We performed an ablation on the GP regression task regarding positional embeddings. The table below supplements our Table 1 from the main paper. In addition to the GP results given in the paper, we trained and deployed our method by removing the positional embeddings of the buffer. Interestingly, for this task we found no statistically significant difference in predictive performance of our buffer with or without positional encoding (and in fact, the scores are nearly identical).
>
> Table R4: Average joint predictive log-density ($\uparrow$) for positional embedding ablation on GP task; mean (SEM). This table reports new ablation results obtained during the rebuttal period.
> |   | TNP-D-AR | TNP-D-Ind  | TNP-ND | TNP-A | Ours, K=16 (fast) | Ours, w/o pos emb, K=16 (fast) |
> |---|---|---|---|---|---|---|
> | GP| 2.568 (0.0195) | 2.222 (0.0223) | 0.798 (0.0821) | 2.243 (0.0184) | 2.513 (0.0195) | 2.511 (0.0194) |
>
> A similar result has been observed in the causal-transformer literature, where autoregressive models without explicit positional encodings still learn robust positional information via the causal mask and network depth (e.g., NoPE) [1–3]. In our work, we started from the simple hypothesis that, since the buffer represents a sequence, adding standard positional embeddings was a natural design choice; however, our ablation together with this prior work suggests they may not be strictly necessary [1–3]. Still, keeping positional embeddings may enable potential future extensions to extrapolate the buffer to larger sizes without necessarily training on them (e.g., via ALiBi [4] or RoPE [5]). A more systematic study of position-free or alternative positional parameterizations of the buffer is therefore an interesting direction for future work. We will add the ablation and discuss the role of positional embeddings in the revised discussion.

---

> ### Author Response · Authors · 2025-11-24
> **Response to Reviewer agRA (Part 3 of 3)**
>
> > **Q1c:** ... and ordering effects when approximating permutation invariance via order-averaging?
> > - Is it possible to quantify order-averaging required to stabilize joint likelihoods with a metric measuring their cost/benefit?"
>
> It is worth noting that order averaging is not unique to our method; it is a requirement for *all* autoregressive methods applied to sets that desire a permutation-invariant estimator of the joint density. Still, it is worth exploring. We conducted additional experiments showing the behavior of our method’s estimator for a varying number of permutations. We present results for the multisensory causal inference problem, which lends itself naturally to this question, as we are using our method precisely to calculate the log-marginal likelihood (LML) of a dataset (averaging over multiple permutations); see results in the two tables below ($\Delta$LML is the difference in LML for the same datasets, for the two different cognitive models we are comparing in the paper). We observe that the RMSE relative to the true joint density decreases at a similar rate to existing autoregressive methods, suggesting no additional "cost" to our approximation compared to baselines, while being significantly faster per permutation.
>
> Table R5: New LML RMSE results with respect to the ground truth ($\downarrow$) on the multisensory causal inference task with varying numbers of sequence samples; results are mean (SEM) over 10 different runs.
> | N. of sequence samples | TNP w/ buffer | TNP-A | TNP-D-AR |
> | :---- | :---- | :---- | :---- |
> | 2 | 3.85 (0.049) | 4.95 (0.062) | 3.35 (0.076) |
> | 4 | 3.66 (0.055) | 4.82 (0.065) | 3.21 (0.047) |
> | 16 | 3.59 (0.024) | 4.75 (0.029) | 3.10 (0.019) |
> | 32 | 3.57 (0.015) | 4.74 (0.017) | 3.10 (0.013) |
> | 64 | 3.56 (0.012) | 4.72 (0.014) | 3.09 (0.011) |
> | 128 | 3.56 (0.004) | 4.74 (0.008) | 3.10 (0.007) |
>
> Table R6: New $\Delta$LML RMSE results with respect to the ground truth ($\downarrow$) on the multisensory causal inference task with varying numbers of sequence samples; results are mean (SEM) over 10 different runs.
> | N. of sequence samples | TNP w/ buffer | TNP-A | TNP-D-AR |
> | :---- | :---- | :---- | :---- |
> | 2 | 3.08 (0.066) | 3.78 (0.143) | 2.90 (0.063) |
> | 4 | 2.79 (0.049) | 3.52 (0.064) | 2.69 (0.049) |
> | 16 | 2.66 (0.017) | 3.35 (0.029) | 2.49 (0.033) |
> | 32 | 2.60 (0.018) | 3.30 (0.029) | 2.45 (0.021) |
> | 64 | 2.60 (0.015) | 3.28 (0.023) | 2.45 (0.008) |
> | 128 | 2.58 (0.010) | 3.28 (0.011) | 2.43 (0.009) |
>
> > How do the authors decide on the size of $(N)$s? Is it there a experiment for empirically showing for large values of N, proposed method has compounding advantages?
>
> The size of $N$ is generally determined by the dataset or task (e.g., number of available weather stations or tabular rows). Our method has compounding advantages for large $N$. The baseline method scales as $O(K(N+K)^2)$, meaning it pays the quadratic cost of $N$ for *every* target generated. Our method pays the $O(N^2)$ encoding cost only once for each $K$ points we predict. Therefore, as $N$ grows, the gap between our runtime and the baseline's runtime widens quadratically. Figure 3 in the existing submission plots these efficiency results as a function of $N$ on the $x$-axis, demonstrating this divergence.
>
> Moreover, as shown with our new results (see earlier Tables in **Q1a**), the relative accuracy of our method improves with large $N$, showing that it’s both faster and accurate relative to the other methods.
>
> **References:**
>
> [1] Haviv et al., *Transformer Language Models without Positional Encodings Still Learn Positional Information*, Findings of EMNLP 2022.
>
> [2] Irie, *Why Are Positional Encodings Nonessential for Deep Autoregressive Transformers? Revisiting a Petroglyph*, Findings of ACL 2025.
>
> [3] Zuo et al., *Position Information Emerges in Causal Transformers Without Positional Encodings via Similarity of Nearby Embeddings*, COLING 2025.
>
> [4] Press et al., *Train Short, Test Long: Attention with Linear Biases Enables Input Length Extrapolation*, ICLR 2022.
>
> [5] Su et al., *RoFormer: Enhanced Transformer with Rotary Position Embedding*, Neurocomputing 2024.

---

### Official Review · Reviewer_Jq2K · 2025-10-31

**Soundness:** 3
**Presentation:** 3
**Contribution:** 3
**Rating:** 8
**Confidence:** 2

**Summary:**

The paper addresses a conflict between two types of models: set-based probabilistic models and autoregressive models. When models (like Neural Processes) try to generate a sequence of predictions (e.g., filling in a signal), they have to re-process all the original data and all the new predictions at every single step. The core idea is to decouple the static context from the dynamic predictions by introducing a new architectural component called the "causal autoregressive buffer". The advantage is (i) on the computational advantage: The expensive $\mathcal{O}(N^2)$ cost (encoding the context) is paid only once at the beginning and (ii) preserved accuracy as shown in the experiments section.

**Strengths:**

1. The paper is well-written and novel. The problem of repeated, expensive re-encoding is visualized in Figure 1, which provides an intuitive understanding of the entire paper's premise. The proposed solution is an elegant architectural fix that decouples the static context from the dynamic predictions, supported by a clear complexity analysis ($\mathcal{O}(N^{2}+NK+K^{2})$).

2. The empirical evidence is convincing. It delivers a speedup (up to 20x) with small additional training cost and without sacrificing predictive accuracy.

**Weaknesses:**

I'm not an expert in this field (neural processes, probabilistic meta-learning, amortized inference), so I'll leave my comments in the following Questions sections.

**Questions:**

1. What happens when the target sequence length M is larger than the buffer size? I saw a discussion on Appendix E.2 and also the limitation in the final discussion section, which suggests that to achieve better performance, we need more computational time. I'm wondering what the final complexity is if including the operation "we evaluate every $K$ targets once and perform AR for $M/K$ steps". Will this still lead to the 20 times speed up? How often does the situation "target count exceeds training bounds of the buffer" happen in real-world analysis?

---

> ### Author Response · Authors · 2025-11-24
> **Response to Reviewer Jq2K (Part 1 of 1)**
>
> We thank the reviewer for the strong score and for finding our work novel, elegant, and the empirical evidence convincing.
>
> > **Weaknesses:** I'm not an expert in this field (neural processes, probabilistic meta-learning, amortized inference), so I'll leave my comments in the following Questions sections.
>
> We appreciate the thoughtful questions, which we answer below.
>
> > **Questions:**
>
> > **Q1:** What happens when the target sequence length M is larger than the buffer size? I saw a discussion on Appendix E.2 and also the limitation in the final discussion section, which suggests that to achieve better performance, we need more computational time. I'm wondering what the final complexity is if including the operation "we evaluate every $K$ targets once and perform AR for $M / K$ steps". Will this still lead to the 20 times speed up?
>
> You are correct that when $M > K$, the buffer fills up, and we must re-encode the context, as shown in Appendix E.2. Specifically, we perform the $\mathcal{O}(N^2)$ context encoding operation once every $K$ steps (re-embedding the points in the buffer as context and clearing the buffer).
>
> However, the relative speedup of roughly $20\times$ (assuming $K = 16$ as in paper; and also note we have new experiments with up to $K = 64$, see Table R1 in Q2 below) is *preserved* even in this setting because the baseline must perform this expensive operation at **every single step**. Our method incurs this cost only every $K$ steps ($M/K$ times total). Therefore, the complexity when $M > K$ scales as: $\mathcal{O}\left(\frac{M}{K} N^2 + MN + MK\right).$ Because the dominant cost ($N^2$) is reduced by a factor of $K$, the computational advantage over the baseline remains consistent regardless of how large $M$ becomes.
>
> > **Q2:** How often does the situation "target count exceeds training bounds of the buffer" happen in real-world analysis?
>
> This is an excellent question, though difficult to quantify due to the vast variety of real-world applications. In practice, the length of target sequences $M$ varies significantly depending on the domain and task at hand. For example, in the tabular domain, joint predictions of all features in a table for a given row will depend on the number of features to predict. The buffer ranges we tested in the paper (up to $K = 32$) comfortably cover small-to-medium size tables. New experiments for this rebuttal on Gaussian process regression tasks show that buffer size can be extended further at training with no loss of performance (up to $K = 64$; see Table R1 below for the result). As another example, using transformer neural processes for planning tasks such as Bayesian optimization (see for example [1-3]) might require to compute fast “rollouts” of future function observation up to a certain horizon $T$. In many such tasks, $T$ is of the order of ~10-30 observations, which is covered by the buffer ranges tested in our paper.
>
> Table R1: Average joint predictive log-density ($\uparrow$) on GP task across target counts up to 64 with number of context ($N$) of 16 and 128, reported as mean (SEM). These new results were obtained using models trained with a maximum buffer size $K$ of 64.
> | Model Configuration | 8 Targets | 16 Targets | 32 Targets | 64 Targets |
> | :--- | ---: | ---: | ---: | ---: |
> | **$N=16$, TNP w/ buffer, $K=1$** | 1.78 (0.04) | 2.00 (0.04) | 2.22 (0.04) | 2.64 (0.03) |
> | **$N=16$, TNP w/ buffer, $K=\text{Targets}$** | 1.76 (0.04) | 1.96 (0.04) | 2.16 (0.04) | 2.58 (0.03) |
> | **$N=16$, TNP-D-AR** | 1.87 (0.04) | 2.07 (0.04) | 2.28 (0.04) | 2.68 (0.04) |
> | **$N=128$, TNP w/ buffer, $K=1$** | 3.53 (0.03) | 3.57 (0.02) | 3.61 (0.02) | 3.66 (0.02) |
> | **$N=128$, TNP w/ buffer, $K=\text{Targets}$** | 3.53 (0.03) | 3.57 (0.02) | 3.60 (0.02) | 3.64 (0.02) |
> | **$N=128$, TNP-D-AR** | 3.51 (0.03) | 3.55 (0.03) | 3.58 (0.02) | 3.63 (0.02) |
>
> In sum, it is hard to predict exactly how often $M$ will exceed the training bounds of the buffer ($K$) in a general sense, but we believe that the ranges shown in the paper already enable users to perform a variety of useful tasks in many domains. Moreover, this variability is exactly why we designed the architecture to be flexible. Whether a task is short ($M \le K$) or requires long-horizon generation ($M > K$), our method does not strictly require the user to manually tune the buffer size. It automatically utilizes the buffer for maximum efficiency and seamlessly switches to the re-encoding regime only when necessary. This ensures the inference algorithm remains efficient regardless of the specific target size of the application.
>
> **References:**
>
> [1] Maraval et al., *End-to-end Meta-Bayesian Optimisation with Transformer Neural Processes*, NeurIPS 2023.
>
> [2] Zhang et al., *PABBO: Preferential Amortized Black-Box Optimization*, ICLR 2025.
>
> [3] Huang et al., *ALINE: Joint Amortization for Bayesian Inference and Active Data Acquisition*, NeurIPS 2025.

---

### Official Review · Reviewer_G5io · 2025-10-31

**Soundness:** 3
**Presentation:** 3
**Contribution:** 2
**Rating:** 4
**Confidence:** 3

**Summary:**

The paper proposes a “causal autoregressive buffer” to accelerate joint inference in transformer-style probabilistic models such as Transformer Neural Processes (TNPs) and Prior-Fitted Networks (PFNs). The idea is: instead of repeatedly re-encoding the entire context set at every autoregressive step, the model encodes the context once, caches it, and then maintains a lightweight causal buffer of generated targets. This is claimed to both (i) avoid recomputing attention over the full growing set, and (ii) still model target-to-target dependencies during sequential rollout. The authors argue this reduces complexity from
$O(K (N+K)^2) \) to \( O(N^2 + NK + K^2) $
and yields up to 20× faster joint sampling while matching predictive quality of stronger autoregressive baselines, across synthetic regression, EEG forecasting, and tabular regression tasks.

**Strengths:**

### 1 Practical inference speedup for probabilistic transformers
The work directly targets a real bottleneck: autoregressive joint inference in set-conditioned probabilistic models is extremely expensive if you have to repeatedly re-encode the entire (context ∪ generated targets) set at every step. The proposed causal autoregressive buffer reduces the need to rerun the full encoder every time by caching the context representation once and then only extending a lightweight causal buffer of generated targets. This reduces the stated complexity from \(O(K (N+K)^2)\) to \(O(N^2 + NK + K^2)\) for K rollout steps, which the authors claim translates into 3×–20× wall-clock speedups in their experiments.
This is a meaningful systems win for anyone who wants to deploy TNP/PFN-style models in real-time or interactive settings.

### 2 Keeps target-to-target dependency modeling (vs. fully independent heads)
A common cheap fallback in meta-regression / neural process style models is to just predict each target independently conditioned on the same context encoding. That is fast but discards correlations across targets.
Here, the buffer still allows each new target to attend causally to previously generated targets in the rollout, so you capture sequential statistical structure across predicted outputs without paying the full re-encode cost. This is especially relevant for tasks like trajectory / time-series forecasting (e.g. EEG forecasting) where temporal consistency matters.
In other words, you get something closer to coherent joint samples instead of just pointwise fits.

### 3 Unification: one model that can act both “independently” and “autoregressively”
The paper proposes a training curriculum in which, during training, some targets are forced to condition only on the static context block (independent prediction mode), and others are allowed to attend to a prefix of previously generated targets through the buffer (autoregressive mode). The same set of parameters is trained to handle both regimes.
If this actually works robustly, it’s attractive from an engineering perspective: you don’t need to maintain two separate heads (one batchable + independent, one fully autoregressive). You get a single model that can cheaply do fast batched prediction or slower-but-more-coherent sequential rollout.

### 4 Conceptual bridge between permutation-invariant context encoders and standard AR decoding
Architecturally, the paper is trying to marry two worlds:
- permutation-invariant / set-encoder style context processing (as in Neural Processes, PFNs, etc.), and
- standard causal decoding with KV caching (as in language models).

**Weaknesses:**

## 2. Major Concerns

### 2.1 Claimed novelty vs. standard KV caching
The proposed “causal autoregressive buffer” is essentially a block-structured attention mask plus caching of keys/values:

- A *static* context block (permutation-invariant encoder over the observed context set) is computed once and cached.
- A *growing* autoregressive buffer stores keys/values for previously generated targets and is updated step by step with a causal mask.
- During generation, new target tokens attend to (a) the frozen context block and (b) previous targets in the buffer, but the buffer cannot “write back” into the context block.

This is very close in spirit to ordinary autoregressive Transformer inference with KV caching: you prefill a prefix once, then incrementally extend the cache with each new token under a causal mask.

The paper repeatedly describes this as a general architectural mechanism that “decouples context encoding from sequential prediction,” but standard decoder-only Transformers already do this: prefix context is cached, new tokens causally attend to the prefix and prior tokens. The only real twist here is that the “prefix” came from a permutation-invariant encoder instead of from plain tokens, and that the mask enforces a hard separation between context and generated targets (“R1–R4”).
Right now, the paper oversells this as conceptually new. It reads more like applying known KV-caching style inference to set-conditioned meta-learners. The authors need to justify why this is more than “cache the context encodings and use a causal mask for the generated block.”

---

### 2.2 Theory is bookkeeping, not analysis of approximation error
The main “theoretical” contribution is a complexity argument: naive autoregressive deployment of a TNP-like model requires re-running attention over the enlarged context-plus-buffer at every step, costing \( O(K (N+K)^2) \), while the buffer method pays \( O(N^2) \) once for the context and then \( O(NK + K^2) \) to roll out K targets.

This is essentially computational accounting, not a theoretical guarantee about inference quality. The paper makes a strong claim that the buffered rollout “preserves model quality,” because each new prediction can still attend to all prior targets via the buffer and to the cached context. But there is no formal analysis of when this approximation matches the behavior of the “true” fully-updated model that would have re-encoded the augmented context set after each new target is added.

In fact, once you stop re-encoding, you’ve thrown away strict permutation invariance over the *union* of context and generated targets. After generation begins, the model is no longer allowed to revisit and re-symmetrize the combined set. The paper acknowledges degradation when the number of autoregressive targets exceeds the buffer size K used during training.
This is a core limitation, but it’s treated as an implementation detail rather than a fundamental modeling gap.

---

### 2.3 Empirical evaluation seems arranged to flatter the method
Most of the storytelling is about runtime: joint sampling speed, likelihood evaluation speed, etc. The new buffer method is reported as 3–20× faster than autoregressive baselines like TNP-A or TNP-D-AR, which repeatedly re-encode context at each step, especially at large N.

Concerns:

- The paper admits their implementation uses heavy engineering (FlashAttention-2, Triton kernels, KV caching), and also says baselines were “optimized beyond their public versions.” But we never see an ablation that isolates the architectural contribution from pure systems tuning. We can’t tell if the 20× number is “new idea” vs. “better CUDA.”
- There is no memory comparison. The buffer maintains per-layer KV caches for both the frozen context block and the autoregressive buffer tokens, which can blow up for long rollouts and large batch sizes. Only wall-clock time is emphasized, which is convenient but incomplete.
- When you actually look at predictive quality (e.g. EEG forecasting), the buffered model with K=16 can underperform the slow full autoregressive baseline (TNP-D-AR), and sometimes is even worse than extremely small-buffer variants (effectively K=1).
  So the tradeoff is not “same accuracy, way faster.” It’s “sometimes noticeably worse, but faster.”

Despite that, the abstract and intro still assert that the method “matches predictive accuracy … while delivering up to 20× faster joint sampling,” which is too strong given these cases.

In short, the evaluation is narrated as “little to no quality loss, huge speedup,” but the actual numbers show a real quality vs. speed tradeoff.

---

### 2.4 Generality claims are speculative and weakly supported
The paper claims broad applicability: Perceiver-like architectures with pseudo-tokens, probabilistic neuroscience modeling, and “tabular foundation models” such as PFN-style inference.

But:

- The Perceiver extension is hypothetical; there is no experiment demonstrating the buffer with learned pseudo-tokens.
- The “tabular foundation model” experiment is actually quite small. The model is trained from scratch on synthetic structural causal model data, then evaluated on a few UCI-like tabular tasks with N=128 context / M=32 targets.
  That’s nowhere near the scale implied by the phrase “foundation model.”
- In those tabular results, “AR w/ buffer (K=32)” performs roughly on par with a standard AR baseline (K=1) and modestly better than fully independent predictions.
  That mostly shows that causal conditioning across targets helps, which is expected. It does not prove that the specific buffering trick is uniquely enabling.

So the claim that this mechanism is a general, broadly applicable inference upgrade feels speculative. The evidence is narrow, and in some cases purely suggestive.

---

### 2.5 Missing baselines / missing ablations
The baselines are primarily TNP variants (TNP-D, TNP-D-AR, TNP-A, etc.).
However, to really argue “our method is necessary,” I would expect at least:

1. **A simple two-tower baseline**
   - Tower A encodes the context set once (frozen).
   - Tower B is a causal decoder over targets that cross-attends into Tower A, with standard KV caching.
   - This is, in spirit, what the proposed model is.
   Without ablations on masking structure and curriculum, we don’t know if the fancy buffer rules (R1–R4) are actually critical or if a trivial cross-attend-decoder would do the same.

2. **A plain autoregressive Transformer treating the context as a prefix**
   - Just linearize the context into a token sequence, feed it as a prefix prompt, and then autoregress over targets exactly like a language model, ignoring permutation invariance entirely.
   - The paper insists that permutation invariance of the context block is crucial, but it never quantifies how much you lose if you drop that constraint and just go full decoder-style.

**Questions:**

see in weaknesses

---

> ### Author Response · Authors · 2025-11-24
> **Response to Reviewer G5io (Part 1 of 5)**
>
> We thank the reviewer for acknowledging the practical inference speedup, the benefits of maintaining target-to-target dependencies, and the practical advantages of a unified model. We appreciate the detailed feedback and address the major concerns below, providing clarifications and new experimental results.
>
> ### W2.1: Claimed novelty vs. standard KV caching
>
> > The proposed 'causal autoregressive buffer' is essentially a block-structured attention mask plus caching of keys/values [...] This is very close in spirit to ordinary autoregressive Transformer inference with KV caching [...] The authors need to justify why this is more than 'cache the context encodings and use a causal mask for the generated block.'
>
> We appreciate the opportunity to clarify this distinction. While our method *utilizes* KV caching, our contribution lies in the **specific architectural design required to enable it efficiently within this class of models (TNP/PFNs)**, which presents unique challenges not addressed by standard transformer implementations.
>
> Simply applying standard KV caching is not feasible due to a fundamental structural conflict:
> * **The Set Constraint:** TNPs/PFNs require bidirectional (non-causal) self-attention within the context set to maintain permutation invariance.
> * **The Sequential Constraint:** Autoregressive generation requires causal masking to capture target dependencies.
>
> Standard KV caching (as used in LLMs) assumes a purely causal sequence. It does not natively support the hybrid state required here: a frozen, bidirectional set-representation acting as a prefix for a causal sequence, trained within a single unified model.
>
> Our proposed method resolves this conflict through our specific masking strategy (requirements R1-R4 in Section 3 of the paper), training curriculum, and inference algorithm. This solution is what allows set-based models to utilize efficient caching without sacrificing the permutation invariance of the context, which is a novel integration in the NP/PFN literature.
>
> ### W2.2: Theory is bookkeeping, not analysis of approximation error
>
> > The main “theoretical” contribution is a complexity argument: naive autoregressive deployment of a TNP-like model requires re-running attention over the enlarged context-plus-buffer at every step, costing $O(K (N+K)^2)$, while the buffer method pays $O(N^2)$ once for the context and then $O(NK + K^2)$ to roll out K targets. This is essentially computational accounting, not a theoretical guarantee about inference quality. [...]
>
> This is correct: our method is more efficient in terms of computational complexity, although it’s important to show that such theoretical efficiency bears in practice (as shown in our paper). We agree that the *efficiency* of our method is per se no guarantee of inference quality. Our demonstration is indeed empirical: we conducted a large variety of experiments that provide strong empirical evidence that our method works well in practice.
>
> > In fact, once you stop re-encoding, you’ve thrown away strict permutation invariance over the union of context and generated targets [...]
>
> We agree that our method treats generated targets as a sequence, and not a set. Our method freezes $\mathcal{C}$ and treats $y_{1:t}$ as an ordered sequence. However, we emphasize the key point that that *no* autoregressive method (including re-encoding generated targets at each step) yields a joint density that is permutation invariant with respect to the targets.
>
> Both approaches rely on sampling (or evaluating) the joint distribution under a particular factorization $p(y_1, \dots, y_M) = \prod_{t=1}^M p(y_t | y_{<t})$. This factorization is inherently order-dependent: generating targets in different orders yields a different joint probability, *regardless of whether the generated targets are re-encoded as a set alongside the context or cached as a sequence*.
>
> Since target permutation-invariance is already broken by the autoregressive factorization itself, the computational cost of "re-symmetrizing" the new generated target at every step is $\mathcal{O}((N + m)^2)$, and yields diminishing returns.
>
> Importantly, this shows that our paper contains results that are potentially unexpected and partly against the common belief that permutation-invariance is needed at every step (although, as we will show below, permutation invariance in the *context* is crucial). Our results confirm that treating the generated targets as a fixed sequence (our buffer) performs almost on par with treating them as a re-encoded set, but with $\mathcal{O}(1)$ complexity.

---

> > ### Author Response · Authors · 2025-11-24
> > **Response to Reviewer G5io (Part 2 of 5)**
> >
> > > **W2.2 (Continued)**
> >
> > > The paper acknowledges degradation when the number of autoregressive targets exceeds the buffer size K used during training. This is a core limitation, but it’s treated as an implementation detail rather than a fundamental modeling gap.
> >
> > We wish to clarify what seems like a crucial misunderstanding. When the number of targets $M$ exceeds the trained buffer size $K$, **the model's predictive quality does not degrade.** As detailed in Appendix E.2, when $M > K$, the model does not fail. We simply empty the buffer, commit the generated points to the context (re-encoding the expanded context), and proceed. The "limitation" is purely computational: the $\mathcal{O}((N+m)^2)$ re-encoding cost, which the baseline pays at every step, is incurred only every $K$ steps by our method. Therefore, the speedup factor is capped by $K$, but the accuracy is preserved.
> >
> > ### W2.3: Empirical evaluation seems arranged to flatter the method
> >
> > We wish to provide evidence that clarifies the fairness of our empirical evaluation and the source of the reported speedups.
> >
> > **1. Isolating Architecture vs. Engineering**
> > > The paper admits their implementation uses heavy engineering [...] But we never see an ablation that isolates the architectural contribution from pure systems tuning. We can’t tell if the 20× number is 'new idea' vs. 'better CUDA.'"
> >
> > We invite the reviewer to check out Appendix C of our paper, as mentioned in the original main text. To summarize, we explicitly isolated these factors in Appendices C.2-C.3 and Figures A4-A6 to ensure a **fair comparison**:
> >
> > * Optimization Parity: We applied the same optimizations (kernels, compilation, KV caching, where applicable) to all baselines. Thus, the main paper results **already** isolate the architectural contribution by holding "systems tuning" constant.
> >
> > * Validating Strong Baselines (Figs A5 & A6): To ensure we were not inflating our numbers, Figures A5 and A6 demonstrate that our optimized baselines are already up to an order of magnitude faster than public code. In the main paper, we compare against the optimized baselines. The reported speedup is *on top* of these highly optimized baselines.
> >
> > * Kernel Ablation (Fig A4): We explicitly ablated the contribution of the custom Triton kernel in Figure A4.
> >
> > * Our Method **enables** the Custom Triton Kernel: We clarify that the custom kernel is only possible *because* of the architectural decoupling. It efficiently computes cross-attention between a batched target stream (size $B$) and a single shared context cache (size $1$). Standard baselines **cannot** use this optimization because they must materialize unique, growing context states for every element of the batch.
> >
> > **2. Memory Comparison**
> > > "There is no memory comparison. The buffer maintains per-layer KV caches... which can blow up for long rollouts and large batch sizes."
> >
> > Thank you for this suggestion. We measured peak VRAM usage (GB) and our method is significantly more efficient than standard autoregressive baselines. As shown in **Table R1**, our method uses up to $6\times$ **less memory** compared to the gold standard TNP-D-AR. This is because:
> >
> > * Our context $\mathcal{C}$ is immutable, so we only need to store **one shared copy** of the context KV-cache shared across all $B$ batch elements.
> > * Baselines must maintain $B$ separate copies of the growing context/target set.
> >
> > The new results in the table alongside an ablation for varying batch sizes will be added to the updated submission.
> >
> > Table R1: New results on peak VRAM comparison (GB) for batch size $B = 256$ (varying number of contexts $N$).
> > | Method | N=32 | N=64 | N=128 | N=256 | N=512 | N=1024 |
> > | :--- | :--- | :--- | :--- | :--- | :--- | :--- |
> > | TNP-D-AR | 0.092 | 0.15 | 0.26 | 0.48 | 0.91 | 1.8 |
> > | TNP-A | 0.12 | 0.17 | 0.29 | 0.50 | 0.94 | 1.8 |
> > | Ours | 0.046 | 0.054 | 0.069 | 0.10 | 0.16 | 0.29 |

---

> > > ### Author Response · Authors · 2025-11-24
> > > **Response to Reviewer G5io (Part 3 of 5)**
> > >
> > > **3. Predictive Quality and Trade-offs**
> > > > "When you actually look at predictive quality (e.g. EEG forecasting), the buffered model with K=16 can underperform the slow full autoregressive baseline (TNP-D-AR)... the evaluation is narrated as 'little to no quality loss, huge speedup,' but the actual numbers show a real quality vs. speed tradeoff."
> > >
> > > Please note that our paper contains 12 distinct quantitative comparisons split across multiple datasets and tasks (GP, Sawtooth, 2 x EEG, 2 x Multisensory modeling, 6 x Tabular Foundation Models). 11 out of 12 show near-parity with the gold-standard autoregressive baselines (now 16 out of 17 including the results from the new experiments for the rebuttal). We believe that achieving a $20\times$ speedup while maintaining parity on the near totality of diverse tasks represents a favorable trade-off and a significant contribution.
> > >
> > >
> > > ### W2.4: Generality claims are speculative and weakly supported
> > >
> > > While our paper already provides a variety of empirical results that support our claims, we are happy to provide further evidence and clarification to support the generality of our approach.
> > >
> > > **1. Perceiver Extension**
> > > > "The Perceiver extension is hypothetical; there is no experiment demonstrating the buffer with learned pseudo-tokens."
> > >
> > > We agree that empirical validation strengthens this claim. We have conducted new experiments training a Perceiver-style latent-bottleneck architecture [1]  with our autoregressive buffer, evaluating it on GP task with target counts of 4 and 16 (**Table R2 and R3**).
> > > Interestingly, the results show that the Perceiver with our AR buffer slightly exceeds the performance of standard autoregressive deployment of the Perceiver architecture. This result is likely due to the fact that the buffer allows the model to *explicitly* represent the recent history of targets, bypassing the compressed representation of the context for immediate short-term dependencies, thus slightly enhancing predictive performance. In the context of pseudo-tokens methods, allowing a temporary “uncompressed” buffer which is easier to attend to is a potential advantage of our approach we had not originally foreseen, and we thank the reviewer for recommending these experiments. These results will be added to the updated submission.
> > >
> > > Table R2: Autoregressive buffer extension on perceiver-style model. Average joint predictive log-density ($\uparrow$) on GP example reported as mean (SEM) for number of targets $M=4$.
> > > | Number of contexts | Perceiver-style Ind | Perceiver-style K=1 AR | Perceiver-style w/buff K=M |
> > > | ----: | :---- | :---- | :---- |
> > > | 8 | \-0.40 (0.02) | \-0.34 (0.02) | \-0.14 (0.02) |
> > > | 16 | 0.13 (0.02) | 0.15 (0.02) | 0.33 (0.02) |
> > > | 32 | 0.46 (0.02) | 0.47 (0.02) | 0.59 (0.02) |
> > > | 64 | 0.63 (0.02) | 0.64 (0.02) | 0.73 (0.02) |
> > > | 128 | 0.73 (0.02) | 0.73 (0.02) | 0.81 (0.02) |
> > >
> > > Table R3: Autoregressive buffer extension on perceiver-style model. Average joint predictive log-density ($\uparrow$) on GP example reported as mean (SEM) for number of targets $M=16$.
> > > | Number of contexts | Perceiver-style Ind | Perceiver-style K=1 AR | Perceiver-style w/buff K=M |
> > > | ----: | :---- | :---- | :---- |
> > > | 8 | \-0.47 (0.01) | \-0.01 (0.02) | 0.51 (0.03) |
> > > | 16 | 0.09 (0.02) | 0.34 (0.02) | 0.86 (0.03) |
> > > | 32 | 0.47 (0.02) | 0.56 (0.02) | 1.06 (0.03) |
> > > | 64 | 0.69 (0.02) | 0.72 (0.02) | 1.19 (0.03) |
> > > | 128 | 0.80 (0.02) | 0.80 (0.02) | 1.24 (0.02) |
> > >
> > >
> > > **2. Tabular Foundation Model Scale**
> > > > The 'tabular foundation model' experiment is actually quite small [...] trained from scratch on synthetic structural causal model data [...] That’s nowhere near the scale implied by the phrase 'foundation model.'
> > >
> > > We respectfully clarify that our experimental training setup aligns with the established methodology in the Prior-Fitted Network (PFN) literature. Our model size and training data (10.24 million synthetic datasets) are comparable to the original TabPFN v1 [2]. We use the term "Foundation Model" consistently with its definition in this specific domain (e.g., TabPFN, TabICL). PFNs are explicitly defined by pre-training on synthetic priors (like data generated from structural causal models, or SCM) to generalize to real-world tasks. We follow the standard paradigm in using synthetic SCM data to train our model [2, 3, 4].

---

> ### Author Response · Authors · 2025-11-24
> **Response to Reviewer G5io (Part 4 of 5)**
>
> **2. Tabular Foundation Model Scale continued**
>
> > "then evaluated on a few UCI-like tabular tasks with N=128 context / M=32 targets"
>
> This is a fair point and we agree that it’s important to demonstrate our method on more and larger tabular datasets. To address this concern, we have expanded the tabular evaluation to include three additional datasets (two time-series datasets, “Tetouan Power” and Jena Climate, and one standard tabular dataset, “California House”) and swept context sizes $N$ from $16$ to $1024$ for all datasets. The results consistently show that the buffered method matches the gold-standard AR strategies across this wider suite.
>
>
> Table R4: Average joint predictive log-density ($\uparrow$) for various tabular foundation model tasks with number of context $N = 16$ reported as mean (SEM). These new $N = 16$ results, together with three additional tasks (Tetouan Power, Jena Climate, California Housing) were run during the rebuttal period.
> | Task | Independent | Standard AR | AR w/ buffer |
> | :--- | :--- | :--- | :--- |
> | **Electric consumption** | | | |
> | Int | 0.38 (0.22) | 0.88 (0.19) | 0.78 (0.19) |
> | For | -2.87 (0.77) | -0.97 (0.52) | -1.00 (0.51) |
> | **Gas Turbine** | | | |
> | Int | -0.65 (0.15) | -0.48 (0.13) | -0.48 (0.13) |
> | For | -1.46 (0.26) | -1.00 (0.18) | -0.98 (0.17) |
> | **Bike Sharing** | | | |
> | Int | 0.84 (0.11) | 1.43 (0.11) | 1.31 (0.10) |
> | For | -0.04 (0.17) | 1.02 (0.13) | 0.94 (0.13) |
> | **Tetouan Power** | | | |
> | Int | -0.59 (0.14) | -0.09 (0.14) | -0.10 (0.15) |
> | For | -4.71 (0.46) | -2.39 (0.22) | -2.41 (0.23) |
> | **Jena Climate** | | | |
> | Int | 0.10 (0.12) | 0.64 (0.12) | 0.55 (0.12) |
> | For | -4.53 (1.07) | -2.15 (0.35) | -2.15 (0.34) |
> | **California House** | | | |
> | Int | -1.31 (0.07) | -1.17 (0.08) | -1.16 (0.09) |
>
> Table R5: Average joint predictive log-density ($\uparrow$) for various tabular foundation model tasks with number of context $N = 1024$ reported as mean (SEM). These new $N = 1024$ results, together with three additional tasks (Tetouan Power, Jena Climate, California Housing) were run during the rebuttal period.
> | Task | Independent | Standard AR | AR w/ buffer |
> | :--- | :--- | :--- | :--- |
> | **Electric consumption** | | | |
> | Int | 1.78 (0.06) | 1.78 (0.06) | 1.79 (0.06) |
> | For | 1.64 (0.18) | 1.70 (0.18) | 1.70 (0.18) |
> | **Gas Turbine** | | | |
> | Int | -0.01 (0.16) | -0.01 (0.16) | -0.01 (0.16) |
> | For | -0.60 (0.29) | -0.47 (0.27) | -0.48 (0.27) |
> | **Bike Sharing** | | | |
> | Int | 2.54 (0.05) | 2.54 (0.05) | 2.53 (0.05) |
> | For | 2.32 (0.07) | 2.40 (0.10) | 2.39 (0.06) |
> | **Tetouan Power** | | | |
> | Int | 0.36 (0.07) | 0.36 (0.07) | 0.36 (0.06) |
> | For | -1.12 (0.35) | -0.08 (0.22) | -0.12 (0.23) |
> | **Jena Climate** | | | |
> | Int | 2.01 (0.06) | 2.01 (0.06) | 2.01 (0.06) |
> | For | 1.56 (0.19) | 1.80 (0.13) | 1.64 (0.16) |
> | **California House** | | | |
> | Int | -0.44 (0.08) | -0.44 (0.08) | -0.44 (0.08) |
>
> ---
>
> > In those tabular results, “AR w/ buffer (K=32)” performs roughly on par with a standard AR baseline (K=1) and modestly better than fully independent predictions. [...] It does not prove that the specific buffering trick is uniquely enabling. So the claim that this mechanism is a general, broadly applicable inference upgrade feels speculative. The evidence is narrow, and in some cases purely suggestive.
>
> We thank the reviewer for highlighting that our buffered method ($K=32$) performs "roughly on par" in terms of predictive quality with the standard slow AR baseline ($K=1$). We would like to emphasize that **this parity is precisely the definition of success for our method**, not a limitation.
>
> * The standard AR baseline ($K=1$) represents the theoretical "gold standard" (full autoregressive conditioning); the objective is to match it. By correctly noting that our method matches the performance of $K=1$, the reviewer has confirmed that our approximation introduces negligible error.
>
> * The "enabling" contribution is not exceeding the accuracy of the gold standard, but making that gold-standard accuracy computationally viable. We achieve the predictive performance of $K=1$ while running orders of magnitude faster.
>
> * Regarding the “modestly better than fully independent predictions” (in the tabular results), this is true for Interpolation (Int) tasks, but not true for Forecasting (For) tasks. In forecasting, AR methods perform *drastically* better than fully independent predictions – as shown also in all other tasks in the paper, where independent predictions are often abysmal.
>
> * Finally, to address the "narrow evidence" concern, we point to the successful deployment across six diverse tasks in the main paper (plus the new Perceiver and expanded tabular results), which demonstrates that this mechanism is broadly applicable.

---

> > ### Author Response · Authors · 2025-11-24
> > **Response to Reviewer G5io (Part 5 of 5)**
> >
> > ### W2.5: Missing baselines / missing ablations
> >
> > **1. The "Two-Tower" Baseline**
> > > A simple two-tower baseline: Tower A encodes the context set once (frozen). Tower B is a causal decoder over targets that cross-attends into Tower A [...] This is, in spirit, what the proposed model is.
> >
> > We respectfully seek clarification on this point, as the architecture described (*"Tower A encodes context [...] Tower B is a causal decoder [...] that cross-attends"*) appears mechanistically similar to the inference mode of our proposed method.
> >
> > * Our method effectively *creates* this Two-Tower structure at inference: the Context Cache acts as the frozen "Tower A", and the Causal Buffer acts as "Tower B".
> >
> > * The masking rules (R1–R4) and curriculum are the precise mechanism required to induce this behavior within a single, parameter-efficient backbone. A naive Two-Tower baseline with separate weights would be less parameter-efficient and exhibit training step times equivalent to or greater than TNP-A, which we showed to be much slower to train than our method. Therefore, we believe our method already represents the optimized version of the baseline requested.
> >
> > **2. The "Plain Autoregressive Transformer"**
> > > A plain autoregressive Transformer treating the context as a prefix [...] ignoring permutation invariance entirely. The paper insists that permutation invariance of the context block is crucial, but it never quantifies how much you lose if you drop that constraint [...]
> >
> > Thank you for this suggestion. This is an ablation of interest to the whole field of (transformer) neural processes [5, 6] and PFNs [2, 4, 7], which have worked for years under the assumption that permutation invariance may be useful when conditioning on *sets*. We implemented the recommended baseline: a standard decoder-only Transformer treating the context as a fixed sequence. As shown in **Tables R6 and R7**, the Transformer decoder performs substantially worse than our method across all context sizes. This confirms that explicitly maintaining the permutation-invariant set structure of the context is critical for performance; simply treating the context set as a sequence is insufficient. We will add these new results to the revised manuscript, surely of relief to researchers in these fields.
> >
> > Table R6: Average joint predictive log-density ($\uparrow$) reported as mean (SEM) on the 1D GP regression task ($M=4$). Results for all methods except the transformer decoder model were reproduced from the original submission.
> > | Number of contexts | TNP-D-AR | TNP w/ buffer, K=1 | TNP w/ buffer, K=Targets | Transformer Decoder |
> > | ----: | :---- | :---- | :---- | :---- |
> > | 8 | 0.77 (0.04) | 0.78 (0.04) | 0.76 (0.04) | 0.09 (0.03) |
> > | 16 | 1.75 (0.04) | 1.75 (0.04) | 1.74 (0.04) | 0.84 (0.03) |
> > | 32 | 2.53 (0.04) | 2.51 (0.04) | 2.50 (0.04) | 1.50 (0.03) |
> > | 64 | 3.09 (0.03) | 3.08 (0.03) | 3.07 (0.03) | 2.05 (0.03) |
> > | 128 | 3.51 (0.03) | 3.51 (0.03) | 3.51 (0.03) | 2.46 (0.02) |
> >
> >
> > Table R7:  Average joint predictive log-density ($\uparrow$) reported as mean (SEM) on the 1D GP regression task ($M=16$). Results for all methods except the transformer decoder model were reproduced from the original submission.
> > | Number of contexts | TNP-D-AR | TNP w/ buffer, K=1 | TNP w/ buffer, K=Targets | Transformer Decoder |
> > | ----: | :---- | :---- | :---- | :---- |
> > | 8 | 1.36 (0.04) | 1.36 (0.04) | 1.26 (0.04) | 0.55 (0.03) |
> > | 16 | 2.07 (0.04) | 2.07 (0.04) | 2.00 (0.04) | 1.10 (0.03) |
> > | 32 | 2.68 (0.04) | 2.67 (0.04) | 2.63 (0.04) | 1.66 (0.03) |
> > | 64 | 3.19 (0.03) | 3.18 (0.03) | 3.16 (0.03) | 2.16 (0.02) |
> > | 128 | 3.55 (0.03) | 3.55 (0.03) | 3.54 (0.03) | 2.50 (0.02) |
> >
> > **References:**
> >
> > [1] Feng et al., *Latent Bottlenecked Attentive Neural Processes*, ICLR 2023.
> >
> > [2] Hollmann et al., *TabPFN: A Transformer That Solves Small Tabular Classification Problems in a Second*, ICLR 2023.
> >
> > [3] Qu et al., *TabICL: A Tabular Foundation Model for In-Context Learning on Large Data*, ICML 2025.
> >
> > [4] Hollmann et al., *Accurate predictions on small data with a tabular foundation model*, Nature 2025.
> >
> > [5] Garnelo et al., *Conditional neural processes*, ICML 2018.
> >
> > [6] Nguyen and Grover., *Transformer Neural Processes: Uncertainty-Aware Meta Learning Via Sequence Modeling*, ICML 2022.
> >
> > [7] Müller et al., *Transformers Can Do Bayesian Inference*, ICLR 2022.

---

### Official Review · Reviewer_jkX4 · 2025-11-02

**Soundness:** 4
**Presentation:** 4
**Contribution:** 3
**Rating:** 8
**Confidence:** 4

**Summary:**

Neural processes are finding applications across a range of areas, including in tabular foundation models like TabPFN and in weather modelling e.g. in Aardvark Weather. In many of these applications, it’s important to be able to model joint predictive distributions that capture the dependencies between target variables at multiple locations e.g. when imputing multiple missing values into a row of a tabular dataset or when predicting precipitation across a region to assess flood risk in weather forecasting.  Autoregressive transformer neural processes are arguably the go-to method for solving such tasks.

This paper makes a neat contribution: it addresses one of the central limitations of autoregressive transformer neural processes and sequence models -- their high computational cost at inference time. It introduces a lightweight causal buffering mechanism that preserves the expressive power and calibration benefits of full autoregression while dramatically reducing computation and latency. This makes autoregressive probabilistic models viable for real-time and large-scale applications, from time-series forecasting to neural operator learning. In doing so, the paper bridges the gap between expressive but slow autoregressive methods and fast but less flexible parallel inference, providing a practical path toward scalable, uncertainty-aware inference in scientific and foundation-model settings.

I liked the paper and would vote for acceptance.

**Strengths:**

This paper has at its core a simple but neat idea that works well. I liked the fact that it had a wealth of experimental results including

1. Synthetic function modelling (Gaussian Process (GP) functions and sawtooth synthetic functions.)
2. EEG forecasting and interpolation including real EEG recordings from 20 human subjects
3. Multisensory causal inference modelling involving simulated and real human behavioural data from a multisensory integration study (Liu et al., 2025).
4. Tabular foundation model experiments
5. Further Efficiency and ablation studies

The results were strong -- the method really works.

The presentation and the writing were very clear and well polished. Figures were very well presented. Generally, I thought it was a well executed piece of work.

**Weaknesses:**

The idea is quite simple which could be viewed as a weakness, but I actually view this as a strength for multiple reasons: it's easy to understand, it's simple to implement, and because of this it could be deployed widely.

The idea might also seem niche as AR-TNPs are not super well-known, but this is also not correct. TNPs in the sim2real setting were rebranded by the impactful TabPFN line of work and the contributions here are directly aligned with this breed of tabular foundation model as the final experiment shows. Moreover, NPs have been deployed in another Nature paper in Aardvark weather, which has also been impactful and which could leverage these results to produce scalable weather forecasts. I expect more and more examples of applied NPs will emerge over the coming years.

**Questions:**

Small point, but I’m not a huge fan of the phrase “joint likelihood” since in many contexts in the paper the primary focus is really producing a "joint predictive density” rather than the likelihood of the parameters need for e.g. learning. Indeed, if you trained a model only using likelihoods derived from univariate predictives, you could then immediately use this to produce joint predictive densities (c.f. the AR CNP paper). So I'd place the emphasis on joint distributions most of the time, rather than on the likelihood functions.

line 49: "However, this breaks the set-based structure”  I think that, at this stage in the paper, this is a bit ambiguous. E.g. in standard mode, adding the generated targets back into the context and recomputing everything is arguably as good as it gets in terms of permutation invariance. Perhaps say “However, this involves significant computational overhead…”

line 161: "In practice, Eq. (2) is not exact for likelihood evaluation as it breaks permutation invariance of the model.” I don’t agree with this. This is really a modelling choice, or something that is determined by the task itself, rather than being wrong or right. You can choose to model the order as latent (and therefore something that you need to average over) or known and fixed (in which case you don’t). For time-series, for example, there is often an implied order over the targets. For spatial data, there might not be.

line 291: you could point out here that discrete diffusion is really an any order AR model in disguise and is very closely related to a NP with a discrete input set (arguably it is an NP) and as such could leverage your ideas.

---

> ### Author Response · Authors · 2025-11-24
> **Response to Reviewer jkX4 (Part 1 of 2)**
>
> Thank you for your thorough and supportive review. We are grateful for your appreciation of the core contribution of our work: bridging the gap between the expressivity of transformer probabilistic models (such as Neural Processes and Prior-Fitted Networks) and the computational inefficiency of autoregressive sampling from such models.
>
> > **Weaknesses:** The idea is quite simple which could be viewed as a weakness, but I actually view this as a strength for multiple reasons: it's easy to understand, it's simple to implement, and because of this it could be deployed widely. The idea might also seem niche as AR-TNPs are not super well-known, but this is also not correct. TNPs in the sim2real setting were rebranded by the impactful TabPFN line of work and the contributions here are directly aligned with this breed of tabular foundation model as the final experiment shows. Moreover, NPs have been deployed in another Nature paper in Aardvark weather, which has also been impactful and which could leverage these results to produce scalable weather forecasts. I expect more and more examples of applied NPs will emerge over the coming years.
>
> Thank you for your insightful recognition of the method's simplicity as a strength (e.g., for adoption and applicability) and not a weakness, and for highlighting the high-impact potential applications. These are exactly the kind of directions we envisioned our work could have a strong impact on down the line. We fully embrace this vision and plan to pursue it in the future.
>
> > **Questions:**
>
> > **Q1:** Small point, but I’m not a huge fan of the phrase “joint likelihood” since in many contexts in the paper the primary focus is really producing a "joint predictive density” rather than the likelihood of the parameters needed for e.g. learning. [...] So I'd place the emphasis on joint distributions most of the time, rather than on the likelihood functions.
>
> Good point and yes, you are absolutely correct. We used the term "likelihood" quite loosely and we agree it is imprecise: it is a (predictive, conditional) density, not strictly a likelihood in the statistical sense. We will update the manuscript to consistently use "joint predictive density" where appropriate.
>
> > **Q2:** line 49: "However, this breaks the set-based structure” I think that, at this stage in the paper, this is a bit ambiguous. E.g. in standard mode, adding the generated targets back into the context and recomputing everything is arguably as good as it gets in terms of permutation invariance. Perhaps say “However, this involves significant computational overhead…”
>
> We agree that this is a fair point; our original phrasing was unclear here. The key point is that standard AR deployment preserves the set property but is computationally inefficient. We will replace the sentence with: *"However, this incurs significant computational overhead: each new prediction must be added back to the conditioning set, triggering a complete re-encoding of the expanded context."*

---

> > ### Author Response · Authors · 2025-11-24
> > **Response to Reviewer jkX4 (Part 2 of 2)**
> >
> > > **Q3:** line 161: "In practice, Eq. (2) is not exact for likelihood evaluation as it breaks permutation invariance of the model.” I don’t agree with this. This is really a modelling choice, or something that is determined by the task itself, rather than being wrong or right. You can choose to model the order as latent (and therefore something that you need to average over) or known and fixed (in which case you don’t). For time-series, for example, there is often an implied order over the targets. For spatial data, there might not be.
> >
> > Thanks for bringing this up as it’s a subtle point. We wrote this comment from the perspective that these models are trained as set-based models, where permutation invariance is a key strength and mathematical requirement (exchangeability). As you correctly point out, the situation is more complex -- some models or tasks might have a valid factorization in one specific order (e.g., Markov chains, ARIMA models). Even so, we might want to be able to decompose these models in *other* orderings to infer missing data for example (as neural processes can do), in which case then we would still need to marginalize over latent orderings.
> >
> > In sum, we agree that in certain circumstances, one may desire to model patterns that do *not* satisfy exchangeability, making this a modeling choice rather than an error, although *in the most general case* we need to marginalize. We will change the footnote to: *"Eq. (2) imposes a specific factorization order. While fixing the order can be a valid modeling choice under certain circumstances, this breaks permutation invariance. In cases where permutation invariance is required, a Monte Carlo approximation can be obtained by averaging over multiple target orderings."*
> >
> > > **Q4:** line 291: you could point out here that discrete diffusion is really an any order AR model in disguise and is very closely related to a NP with a discrete input set (arguably it is an NP) and as such could leverage your ideas.
> >
> > Thank you for this insight. We agree and are very excited about the relationship between masked discrete diffusion models and using neural processes in an autoregressive fashion. We share your view that NPs are related to many other paradigms, a perspective that is often overlooked in the current literature. We plan to expand upon this in the Related Work section of the updated manuscript.

---

### Author Response · Authors · 2025-11-24
**Global Response: Summary of Rebuttal Experiments and Upcoming PDF Update**

We thank all reviewers for their constructive feedback and for highlighting the core strengths of our work: the significant efficiency improvements it offers for set-based transformer probabilistic models, its simplicity and wide applicability, and its potential for use within high-impact applications like weather forecasting and tabular foundation models.

During the rebuttal period, we conducted additional experiments to address the specific questions raised. The additional results reinforce the validity and robustness of our proposed method:


* **Scalability (Large $N$ & $K$):** We extended evaluations to large contexts ($N=1024$) and larger buffers ($K=64$). Results confirm that our method’s relative performance *improves* as context grows, and that performance remains robust as buffer size increases.
* **Memory Efficiency:** We provided a peak VRAM analysis showing our method uses up to **$6\times$ less memory** than standard autoregressive baselines.
* **Generality:** We deployed our buffer on a Perceiver-style latent architecture (outperforming standard AR) and expanded our tabular evaluation to three additional datasets, matching gold-standard autoregressive baselines across the board.
* **Ablations:** We implemented a Transformer decoder baseline (showing that maintaining set-based structure is critical for performance), performed positional embedding ablations, and quantified the sensitivity to the number of target permutations that we average over, showing that behaviour is similar to autoregressive baselines.

We are currently synthesizing these new results (tables, figures, and discussions) and revisions into the manuscript. We will upload a single joint update to the submission PDF in the next couple of days to reflect all these improvements.

We are grateful for the reviewers engagement, which has significantly strengthened the paper. Based on these positive outcomes and clarifications, we look forward to your re-evaluation of our work.

---

> ### Author Response · Authors · 2025-11-27
> **Updated version of the submission PDF**
>
> We have updated the submission's PDF to reflect the additions to the paper based on the reviewers' comments, specifically the changes that we agreed to make in our response to the reviewers' comments. Core changes, or changes in direct response to a reviewer, are marked in red. We kindly note that we made a few other minor changes to improve the structure, formatting, and layout of the document, which have not been marked in red.
>
> We look forward to hearing your comments on our responses to your reviews and the updated PDF.

---

### Author Response · Authors · 2025-12-03
**Message to new Area Chair**

Thank you for taking over as new Area Chair for our paper. Our paper proposes an efficient method for autoregressive (AR) sampling from set-based transformer models (TNPs/PFNs), achieving $\sim 20\times$ speedup while closely matching gold-standard predictive performance.

We would like to briefly summarize how our revised submission and rebuttal address the reviewers’ comments. Two reviewers (**jkX4**, **Jq2K**) expressed strong support, with minor suggestions. For the more critical reviews (**G5io**, **agRA**), we conducted substantial new experiments and clarifications. All changes have been incorporated into the updated PDF. Below, we provide a short reviewer-by-reviewer summary for your convenience.

---

**Reviewer jkX4 (score 8, conf. 4)**

Reviewer *jkX4* expressed strong support for the paper, with minor suggestions about specific statements or terminology in the paper for improved clarity and correctness. We addressed all points in our updated submission PDF.

---

**Reviewer G5io (score 4, conf. 3)**

In our response to Reviewer *G5io*, we conducted numerous new experimental analyses during the rebuttal to address this reviewer's concerns. We also clarified a few misunderstandings of the method and points already addressed in the original submission. We encourage the AC to consult the full review and rebuttal to form a complete picture.

> "Novelty vs. standard KV caching"

Standard KV caching assumes a purely causal sequence and cannot be directly applied to TNPs/PFNs, which require bidirectional attention over the context set. Our contribution is the masking strategy that resolves this conflict, enabling efficient caching within set-based models.

> "Degradation when $M > K$"

This is a misunderstanding. When $M > K$, predictive quality is unaffected; only the speedup factor is bounded (re-encoding occurs every $K$ steps rather than every step). New experiments (Table R1) confirm stable performance.

> "Cannot isolate architecture vs. engineering"

This analysis exists in the original submission (Appendix C, Figures A4-A6, referenced in Section 4). To ensure fair comparison, we optimized baselines 10$\times$ over public code; reported speedups are on top of these strong baselines.

> "No memory comparison"

We provide new results in the updated submission (Figure 3, bottom left; Table R1). Peak VRAM analysis shows that our method uses up to 84% less memory than other autoregressive methods such as TNP-D-AR and TNP-A.

> "Perceiver extension is hypothetical"

We implemented and evaluated a Perceiver-style architecture (new tables R2-R3). Perceiver+our buffer performs favorably compared to standard Perceiver-AR on GP tasks.

> "Tabular experiments too small"

Our setup (10.24M synthetic datasets) matches TabPFN v1 training and standard PFN methodology. We expanded evaluations to 3 additional datasets and a variety of context sizes, up to larger context sizes ($N=1024$). Results consistently match gold-standard AR strategy.

> "Quality vs. speed tradeoff"

11 out of 12 original comparisons (16 of 17 with the new experiments) show near-parity with the gold standard; this supports our overall claims (see also Figure 3, center-bottom panel in the revised paper).

> "Two-Tower baseline missing"

Our contribution is precisely the masking strategy and training curriculum that enable this Two-Tower behavior efficiently within a single model -- this is what the reviewer's suggested baseline would require.

> "Decoder-only baseline missing"

We add new tables R6-R7 to the submission PDF. We implemented this, and the decoder performs substantially worse, confirming that maintaining set structure in the context is essential, not merely convenient.

---

**Reviewer Jq2K (score 8, conf. 2)**

Reviewer *Jq2K* expressed strong support for the paper. Their main question concerned behavior when the number of targets $M$ exceeds the trained buffer size $K$. We clarified that when $M > K$, the model re-encodes only every $K$ steps (compared to every step for baselines). This limits the speedup to a factor of $K$ but does not affect predictive performance, which remains on par with gold-standard autoregressive methods. New experiments (Table R1) confirm stable performance with buffer sizes up to $K=64$.

---

**Reviewer agRA (score 6, conf. 3)**

Reviewer *agRA* requested experiments on larger buffer sizes ($K$), larger contexts ($N$), positional embedding ablations, and order-averaging sensitivity. We addressed all four: extended to (1) $K=64$ and (2) $N=1024$ with no degradation (new Table R1); (3) positional embedding ablation shows no significant effect on a simple GP task (new Table R4); and (4) order-averaging analysis confirms convergence rates match baselines while being faster per permutation (new Tables R5-R6).

---

### Meta-Review · Area_Chair_kPPM · 2026-01-02

**Summary:**

This paper proposes a method to obtained joint predictive distributions in the context of transformer-based models for amortized probabilistic inference. For this, the authors introduce a causal autoregressive buffer. The approach decouples context encoding from updating the conditioning set. The model processes the context once and caches it, while a dynamic buffer captures target dependencies: as targets are incorporated, they enter the buffer and attend to both the cached context and previously buffered targets. This enables efficient batched autoregressive generation and one-pass joint predictive density evaluation. Training integrates set-based and autoregressive modes at minimal additional cost. Across synthetic functions, EEG signals, cognitive models, and tabular data, the method matches the predictive accuracy of strong baselines, but it is 20 times faster at joint sampling. Most of the reviewers agree that this is a nice submission that will receive the attention of the community. They indicate that the proposed idea is simple, but works well, supported by extensive experiments. Moreover, the presentation and the writing were very clear and well polished. Reviewers also identified minor weaknesses such as that the proposed method is very close in spirit to ordinary autoregressive transformer inference with KV caching, or that the experiments, they are rather limited to moderate sizes of K and small/medium-sized contexts, or a missing analysis on the order sensitivity. Some reviewers also indicate that the paper generality claims are speculative and weakly supported (partially addressed in the rebuttal) and that there are missing baselines and ablation studies (partially addressed in the rebuttal). In spite of this, I think this is an interesting paper that will receive the attention of the community.

**Reviewer Concerns:**

The rebuttal successfully addressed most of the minor concerns raised by reviewers. Clarifications were provided on terminology, computational overhead, and new experiments resolved minor questions about the proposed method. Missing baselines were partly addressed through ablation studies, since the authors argued their design includes some of the suggested approaches as particular steps.

**Reviewer Scores:**

It is likely that most of the reviewers have kept their scores, but reviewer G5io is likely to have increased its score.

---

### Decision · Program_Chairs · 2026-01-26

Accept (Poster)